

# Topological and quantum critical properties of the interacting Majorana chain model

## Natalia Chepiga[1] and Nicolas Laflorencie[2]

**1** Kavli Institute of Nanoscience, Delft University of Technology,
Lorentzweg 1, 2628 CJ Delft, The Netherlands
**2** Laboratoire de Physique Théorique, Université de Toulouse, CNRS, UPS, France

## Abstract

We study Majorana chain with the shortest possible interaction term and in the presence of hopping alternation. When formulated in terms of spins the model corresponds to the transverse field Ising model with nearest-neighbor transverse and next-nearest-neighbor longitudinal repulsion. The phase diagram obtained with extensive DMRG simulations is very rich and contains six phases. Four gapped phases include paramagnetic, period-2 with broken translation symmetry, $\mathbb{Z}_2$ with broken parity symmetry and the period-2-$\mathbb{Z}_2$ phase with both symmetries broken. In addition there are two floating phases: gapless and critical Luttinger liquid with incommensurate correlations, and with an additional spontaneously broken $\mathbb{Z}_2$ symmetry in one of them. By analyzing an extended phase diagram we demonstrate that, in contrast with a common belief, the Luttinger liquid phase along the self-dual critical line terminates at a weaker interaction strength than the end point of the Ising critical line that we find to be in the tri-critical Ising universality class. We also show that none of these two points is a Lifshitz point terminating the incommensurability. In addition, we analyzed topological properties through Majorana zero modes emergent in the two topological phases, with and without incommensurability. In the weak interaction regime, a self-consistent mean-field treatment provides a remarkable accuracy for the description of the spectral pairing and the parity switches induced by the interaction.

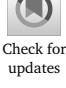

# 1 Introduction

## 1.1 Generalities

Models of strongly-correlated low-dimensional systems attracted a lot of attention in the past decades [1]. Frustrated spin chains, models of spinless fermions and hard-boson models have been studied intensely over the years and have lead to a number of fascinating results. One of them is the emergence of Majorana edge states predicted by Kitaev [2] in a chain of spinless fermions - an effective model of p-wave superconductors. This theory predictions stimulated impressive experimental activity [3–9] motivated by the potential usage of Majorana zero modes in quantum computing [10–12]. By construction the Kitaev chain describes non-interacting Majorana fermions, therefore shortly after the model has been proposed it has also been extended to include interactions of various forms between the Majorana fermions [13–19]. Interacting Majorana fermions have also been theoretically studied in the presence of quenched disorder [20–26], in particular more recently in the context of many-body localization physics at high energy [27–31].

## 1.2 The interacting Kitaev-Majorana chain model

Here we focus on low-energy properties for the following disorder-free microscopic spin Hamiltonian on a one-dimensional (1D) chain:

$$\mathcal{H} = \sum_j \left( J \sigma_j^x \sigma_{j+1}^x - h \sigma_j^z + g_z \sigma_j^z \sigma_{j+1}^z + g_x \sigma_j^x \sigma_{j+2}^x \right), \tag{1}$$

where $\sigma_j^{x,z}$ are Pauli matrices on site $j$. By means of Jordan-Wigner transformation this model can be rewritten in terms of interacting (Dirac) fermions (see Appendix A.2)

$$\begin{aligned}
\mathcal{H} = \sum_j \Big[ & t \left( c_j^\dagger c_{j+1} + \text{h.c.} \right) + \Delta \left( c_j^\dagger c_{j+1}^\dagger + \text{h.c.} \right) + 2h n_j \Big] \\
& + g_z \sum_j \left( 1 - 2n_j \right) \left( 1 - 2n_{j+1} \right) \\
& + g_x \sum_j \left( c_j^\dagger - c_j \right) \left( 1 - 2n_{j+1} \right) \left( c_{j+2}^\dagger + c_{j+2} \right),
\end{aligned} \tag{2}$$

with equal hopping and pairing terms $t = \Delta = J$, and interaction strengths $g_z$ and $g_x$. When $g_{x,z} = 0$ we immediately recognize the celebrated transverse-field Ising (TFI) chain model [32, 33] in Eq. (1), equivalent to the so-called Kitaev chain model as given by the first line of Eq. (2). The two other terms encode the interactions: the second line a nearest-neighbor density-density repulsion for $g_z > 0$ (attraction for $g_z < 0$), while the last one can be understood as a density-assisted hopping and pairing term at second neighbor distance.

This form of interaction takes a more symmetric expression in the Majorana language (see Appendix A.2)

$$\mathcal{H} = -\mathrm{i} \sum_j \left( J b_j a_{j+1} - h a_j b_j \right) - \sum_j \left( g_z a_j b_j a_{j+1} b_{j+1} + g_x b_j a_{j+1} b_{j+1} a_{j+2} \right), \tag{3}$$

where we have introduced two Majorana fermions at each site: $a_j = c_j^\dagger + c_j$ and $b_j = \mathrm{i}\left( c_j^\dagger - c_j \right)$. A sketch of the model is given in Fig. 1 for both spin and Majorana representations. One can easily see that tuning the transverse field of the Ising model in Eq. (1) introduces a bond alternation of the kinetic term in the Majorana chain, see Fig. 1.

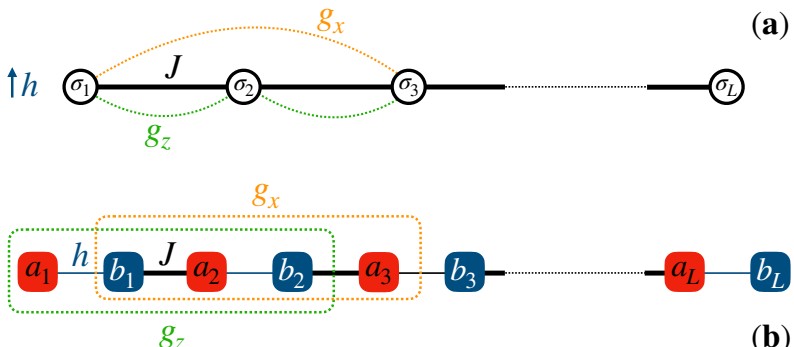

Figure 1: Schematic picture for the spin Hamiltonian (a), and its Majorana fermion representation in (b), both with open boundary conditions. In this work, we fix $J = 1$ and $g_x = g_z = g$.

Various types of interactions have been studied over the past decade, but clearly most of the attention has been devoted to models with density-density interactions $g_z \neq 0$ [15, 18, 19, 34–41]. In particular, the emergence of a stable Luttinger liquid phase [19], the eight-vertex criticality [41] and the exact disorder line [18] have been reported. In contrast, models with $g_x \neq 0$ have not retained much consideration, except a few cases involving correlated hopping of fermions, as studied recently in Refs. [42–44].

In this paper we consider symmetric interactions $g_z = g_x = g$, which display a Kramers-Wannier self-duality, see Appendix A.1. In the rest of the paper we will set $J = 1$ and focus on the repulsive $g > 0$ regime, leaving the attractive case for further studies. In Refs. [16, 17], using field-theory and density matrix renormalization group (DMRG) methods Rahmani *et al.* have achieved a detailed study of symmetric interacting problem for the special case $J = h$. Let us briefly summarize their results for the repulsive situation that we want to consider in this work: For $g \lesssim 0.28$ the system can be described by the Ising critical theory; when $0.28 \lesssim g \lesssim 2.86$ it was suggested that the effective critical theory can be characterized by a central charge $c = 3/2$ and corresponds to the combination of the Ising and the Luttinger liquid criticalities; the latter is accompanied by an emergent $U(1)$ symmetry in agreement with the later analysis by Verresen *et al.* [19]. The point $g \approx 2.86$ has been identified as a generalized commensurate-incommensurate transition beyond which the system is gaped with four-fold degenerate ground-state.

The goal of the present paper is to study an extended phase diagram based on which we rediscuss some of the conclusions drawn in the previous works. Building on DMRG simulations [45–49], we map the global phase diagram of the extended interacting Kitaev-Majorana chain model Eq. (1) with $g = g_x = g_z$ as a function of a transverse field $h$ and interaction strength $g$, see Fig. 2. We perform simulations with two-site DMRG on systems with up to $N = 2000$ sites with open boundary conditions (OBC) and up to $N = 200$ with periodic boundary conditions (PBC). We achieve the convergence by performing up to 8 sweeps keeping up to 3000 states and discarding singular values below $10^{-8}$. When using OBC we either fix them by polarizing the edge spins along the chosen direction or keep them free. The former is used to compute Friedel oscillations profiles in the floating phases and to study quantum phase transitions. The latter is chosen when we study the emergence of Majorana edge states and associated exact zero modes. We also complement the DMRG simulations using a self-consistent Hartree-Fock treatment of the problem in the limit of weak interaction, see Appendix C for some details.

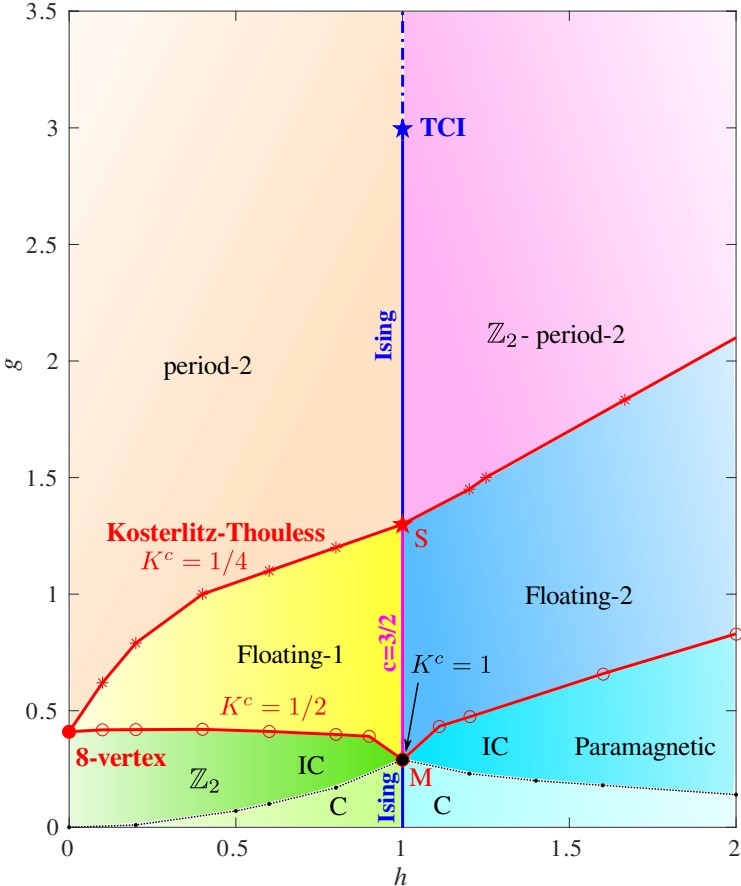

Figure 2: Phase diagram of the interacting Majorana chain model Eq. (4) as a function of coupling constants $g$ and $h$, obtained from DMRG simulations. It contains four gapped phases: $\mathbb{Z}_2$, paramagnetic, period-2 and $\mathbb{Z}_2$-period-2; and two critical floating phases in Luttinger liquid universality class. See main text, Section 2.1 for a brief description of each phase. The model is self-dual for $h \to 1/h$ and $g \to g/h$. The floating phases are separated from the gapped ones by the Kosterlitz-Thouless transitions (red lines) with indicated critical values of the Luttinger liquid parameters $K^c$. The multicritical point at $h = 0$ and $g \approx 0.4105$ is in the eight-vertex universality class. Blue lines are Ising transition that terminates at $g \approx 3$ with the tri-critical Ising point (blue star). For $0.29 \lesssim g \lesssim 1.3$ the Ising transition is superposed with the Luttinger liquid phase resulting in a critical line with central charge $c = 3/2$. Dotted black line states for the disorder line above which the dominant wave-vector is incommensurate.

## 1.3 Paper outline

The rest of the paper is organized as follows. In section 2 we overview the phase diagram briefly discussing the properties of each phase and the nature of the quantum phase transitions between them. In section 3 we discuss in more details the floating phases - Luttinger liquid phases with incommensurate correlations and locate the boundaries of these phases that corresponds to two Kosterlitz-Thouless phase transitions. We then discuss the multicritical point along $h = 0$ line that belongs to the universality class of the eight-vertex model. Equipped with the understanding of the extended phase diagram we revise the nature of the critical lines along $h = 1$ in the section 4. In particular we will provide numerical evidence that there is no generalize commensurate-incommensurate transition predicted in Ref. [17],

the Ising transition persist beyond the Luttinger liquid phase and terminates at the tri-critical Ising point. In section 5 we discuss the Majorana zero modes that appear in the $\mathbb{Z}_2$ regimes, with or without incommensurability. DMRG results are successfully compared with a self-consistent Hartree-Fock treatment in the limit of weak interaction. We summarize our results and put them in perspective in section 6.

## 2 Main results and phase diagram

We investigate the phase diagram of the interacting Majorana chain model defined by Eq. (1) with $g_x = g_z = g$ and $J = 1$ fixed, as a function of the transverse field $h$ and the interaction coupling constant $g$, as rewritten here

$$\mathcal{H} = \sum_j \left[ \sigma_j^x \sigma_{j+1}^x - h\sigma_j^z + g\left( \sigma_j^z \sigma_{j+1}^z + \sigma_j^x \sigma_{j+2}^x \right) \right]. \tag{4}$$

The phase diagram obtained from extensive DMRG calculations is presented in Fig. 2 and consists of six main phases. Below we provide a short summary of our results, describing the different regimes in Sec. 2.1 and the associated phase transitions and (multi) critical points in Sec. 2.2. More details and extended discussions are provided in the corresponding sections of the paper.

### 2.1 Different phases

Before proceeding to a list of various phases let us point out a very important property of the phase diagram. The model defined by Eq. (1) up to boundary terms obeys the Kramers-Wannier self-duality (see Appendix A.1) with $h \to 1/h$ and $g \to g/h$. It immediately defines $h = 1$ as a very special line in the phase diagram where the transitions between each pair of dual phases take place. It turns out that this critical line is associated to a spontaneous parity symmetry breaking for one of the side of the transition, depending on the interaction strength.

#### 2.1.1 Parity ($\mathbb{Z}_2$) broken antiferromagnetic (topological) order

Realized for small $g$ and $h < 1$ with spontaneously broken parity, this gapped phase is topologically non-trivial with a two-fold (parity) degeneracy with emergent Majorana edge states. Note that this phase has no topological interest for the (Ising) spin degrees of freedom for which the so-called "topological phase" there boils down to a more conventional magnetic order.

- $\mathbb{Z}_2$-C: Commensurate region of the $\mathbb{Z}_2$ phase with zero modes showing a vanishing (parity) gap, exponentially suppressed with the system size.

- $\mathbb{Z}_2$-IC: A region of the $\mathbb{Z}_2$ phase with incommensurate short-range order and with zero modes that due to incommensurability are exact at some points even for the finite length of the chain.

#### 2.1.2 Disordered paramagnetic phase

This gapped phase occurs at small $g$ and $h > 1$. The ground-state is short-ranged correlated, and corresponds to all spins polarized along the field $h$.

- Paramagnetic-C: Paramagnet with commensurate short-range order

- Paramagnetic-IC: Paramagnet with incommensurate short-range order showing the same incommensurate vector as its dual phase $\mathbb{Z}_2$-IC.

### 2.1.3 Floating phases

These are gapless Luttinger liquid phases, with quasi-long-range incommensurate order. The corresponding critical theory is characterized by a central charge $c = 1$. The floating phases are stabilized by an emergent $U(1)$ symmetry [17, 19]. In the Floating-2 phase realized for $h > 1$ the parity symmetry $\mathbb{Z}_2$ is spontaneously broken.

### 2.1.4 Period-2 orders

- Period-2 (for $h < 1$): Gapped phase with spontaneously broken translation symmetry and a two-fold degenerate ground-state. The phase is characterized by an antiferromagnetic order along $z$ with $|\langle \sigma_i^z - \sigma_{i+1}^z \rangle| \neq 0$, and on top of it there is an incommensurate short-range order.

- $\mathbb{Z}_2$-period-2 (for $h > 1$): Gapped phase with spontaneously broken translation and parity symmetries. For chains with even number of sites there are Majorana zero modes.

Both phases, at least big portions of them, have incommensurate short-range order, however the dominant wave-vector is very close to the commensurate value and, as we will see, approaches it with a zero slope. It means that for strong repulsion we cannot exclude the existence of the disorder line beyond which the correlations are commensurate. It also implies that any finite-size result pointing towards commensurate wave-vector should be treated with caution - it might take much larger system to detect a presence of incommensurability.

## 2.2 Phase transitions and multicritical points

The phase diagram contains a wide variety of quantum phase transitions:

- **Ising transition:** Quantum phase transition between the $\mathbb{Z}_2$ and the paramagnetic phases is in the Ising universality class along $h = 1$ and for $g \lesssim 0.29$.

- **Ising+LL:** Inside the floating phase Ising transition is superposed with the Luttinger liquid phase resulting in a critical line characterized by the central charge $c = 3/2$. This transition separates two critical floating phases one of which has broken $\mathbb{Z}_2$ symmetry. According to our results Luttinger liquid terminates at $g \approx 1.3$ marked in Fig. 2 with S, for much smaller interaction strength than suggested in Ref. [17].

- **Second Ising transition:** The Ising critical line continues beyond the point S and terminates around $g \approx 3$.

- **Tricritical Ising point:** The location of the end point of the Ising critical line agrees within the error-bars with the location of the generalized commensurate-incommensurate transition reported in Ref. [17]. However, we arrived to a different conclusion regarding the nature of this end point: in Sec. 4 we provide numerical evidences that *(i)* the end point is in the tri-critical Ising (TCI) universality class, *(ii)* the Luttinger liquid terminates at much weaker interaction strength and does not affect the nature of the end point, and *(iii)* the incommensurate short-range correlations persist beyond this end point which is not a Lifshitz point separating commensurate and incommensurate regimes.

- **Kosterlitz-Thouless transitions:** The Luttinger liquid phase is stable against superconducting instability for $K < 1/2$ and against broken translation symmetry for $K > 1/4$. When the Luttinger liquid exponent reaches these critical values, the system undergoes Kosterlitz-Thouless transition into the corresponding gapped phase. Note that in

the period-2 phase short-range order remains incommensurate and the transition belongs to the Kosterlitz-Thouless universality class by contrast to the Pokrovsky-Talapov commensurate-incommensurate transition realized in a related model that takes place at the same critical value of $K^c = 1/4$ [19, 41].

- **Multi-critical point M:** The pairing operator becomes relevant and lead to a gapped $\mathbb{Z}_2$ phase for $K > 1/2$. The spin-flip operation always creates a pair of domain walls in a paramagentic phase and thus has the same critical value $K^c = 1/2$. However, along the Ising critical line neither the pairing nor the spin-flip becomes a relevant perturbation and the Kosterlitz-Thouless transition to the floating phase takes place at $K^c = 1$ that corresponds to free fermions. Two disorder lines (dotted black) separating commensurate and incommensurate regions of paramagnetic and $\mathbb{Z}_2$ phases terminate at this point.

- **Eight-vertex critical point:** The multi-critical point located at $g \approx 0.41$ and $h = 0$ is in the universality class of the eight-vertex model [15, 41, 50, 51].

## 3 Floating phases

### 3.1 The boundaries of the floating phase

Let us first focus on the $h < 1$ side of the phase diagram. The stability of the Luttinger liquid phase against superconducting pairing and spontaneous translation symmetry breaking due to nearest-neighbor repulsion has been studied recently in Ref. [19]. It has been argued that due to an emergent U(1) symmetry the Luttinger liquid phase is stable when the Luttinger parameter lies within the interval $1/4 < K < 1/2$. For $K > 1/2$ the pairing term becomes relevant; for $K < 1/4$ the phase is unstable with respect to spontaneously broken translation symmetry.

By choosing open boundary conditions we explicitly break translation symmetry and therefore can observe the floating phase with Friedel oscillations of local magnetization. In order to reduce log-corrections we polarize first and last spins with strong boundary field. Then according to the boundary conformal field theory, Friedel oscillations on a finite-size system behave as [52]

$$\langle \sigma_i^z \rangle \propto \frac{\cos(qj)}{[(N/\pi)\sin(\pi i/N)]^K} \, , \tag{5}$$

where $K$ is the corresponding scaling dimension that for $\sigma_i^z$ operator (equivalent to the local density operator) coincides with the Luttinger liquid parameter, $q$ is the incommensurate wave-vector.[1] By fitting Friedel oscillations one can get an extremely accurate estimate of both, the Luttinger parameter $K$ and the incommensurate wave-vector $q$. Examples of such fits are presented in Fig. 3 where the uniform part of the spin density $\overline{\langle \sigma_i^z \rangle}$ has been subtracted, and the fitting window is cted to avoid edge effects.

By keeping track of the Luttinger liquid exponent $K$ we can now locate the boundaries of the Floating-1 phase as shown in Fig. 4 (a). Note that due to exponentially slow opening of the gap at the Kosterlitz-Thouless transition it is possible to extract an effective critical exponent $K$ on both sides of the transitions. We therefore associate the location of the transitions with the crossing points of the obtained curve with the corresponding field theory predictions. An important observation is that the wave-vector $q$ is incommensurate on both sides of the floating phase. Therefore the transition to the period-2 phase with spontaneously broken translation

---

[1]Similar results can be obtained for the local pairing term $\langle \sigma_j^x \sigma_{j+1}^x \rangle$ under appropriate boundary conditions.

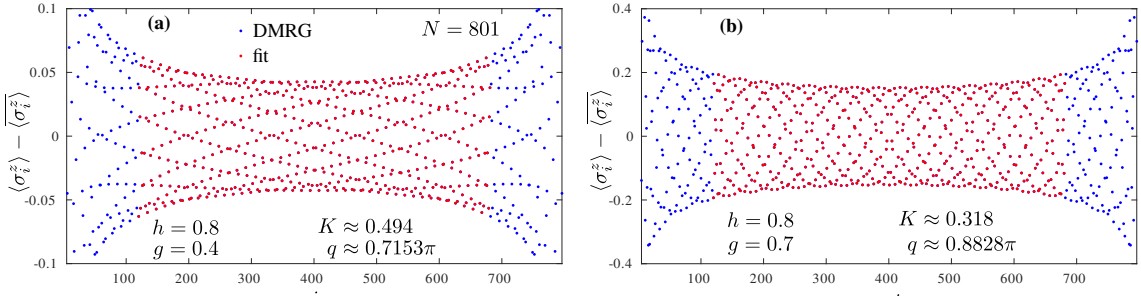

Figure 3: Examples of the Friedel oscillations inside the Floating-1 phase obtained on a finite-size system with $N = 801^2$ sites with polarized boundary conditions. Blue points are DMRG data, red points are the result of the fit with Eq. (5) (blue dots are completely hidden under red ones). Note that the uniform part of the spin density $\overline{\langle \sigma_i^z \rangle}$ has been subtracted, and the fitting window is restricted to the range $i \in [121, 680]$ in order to avoid edge effects.

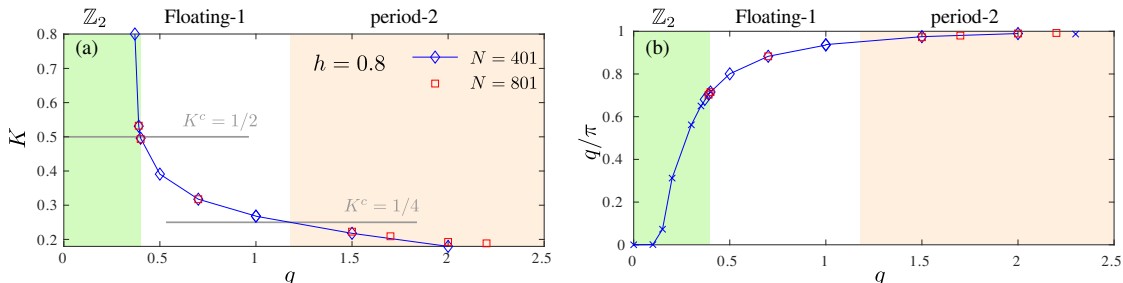

Figure 4: (a) Luttinger liquid parameter $K$ and (b) incommensurate wave-vector $q$ along the vertical cut at $h = 0.8$. The system is in the Luttinger liquid phase for $1/4 < K < 1/2$ (white region). It is clear that the wave-vector remains incommensurate on both sides of the floating phase. Blue diamonds and red squares are results of the Friedel oscillations similar to those presented in Fig. 3; blue crosses state for the wave-vector extracted from the density-density correlation inside the gapped phases.

symmetry is also in the Kosterlitz-Thouless universality class in contrast to the Pokrovsky-Talapov commensurate-incommensurate transition to the period-2 phase with commensurate long- and short-range orders [41].

Relying on the Kramers-Wanier duality, the location of the Kosterlitz-Thouless transition for $h > 1$ can be deduced from the results obtained for $h < 1$. These results are in excellent agreement with independent calculations performed for $h > 1$. In the paramagnetic phase elementary excitations - spin flips - create domain walls in pairs and thus lead to the same value of the critical Luttinger liquid parameter $K = 1/2$. For strong interaction $g$ the same symmetry is broken (the translation) and thus, very naturally, the critical parameter is the same $K = 1/4$.

## 3.2 Floating-1 vs Floating-2

Let us now focus on the similarities and differences between the two floating phases. Because of the self-duality of the model one might expect similar scaling the $z$ component of

---

[2]Odd number of sites is chosen to realize symmetric boundary conditions in the period-2 phase with antiferromagnetic order along $\sigma^z$. Inside the floating phases the even-odd effect is relaxed due to incommensurate wave-vector $q$ but whenever we polarize edges in the longitudinal direction we always take the odd number of sites for consistency.

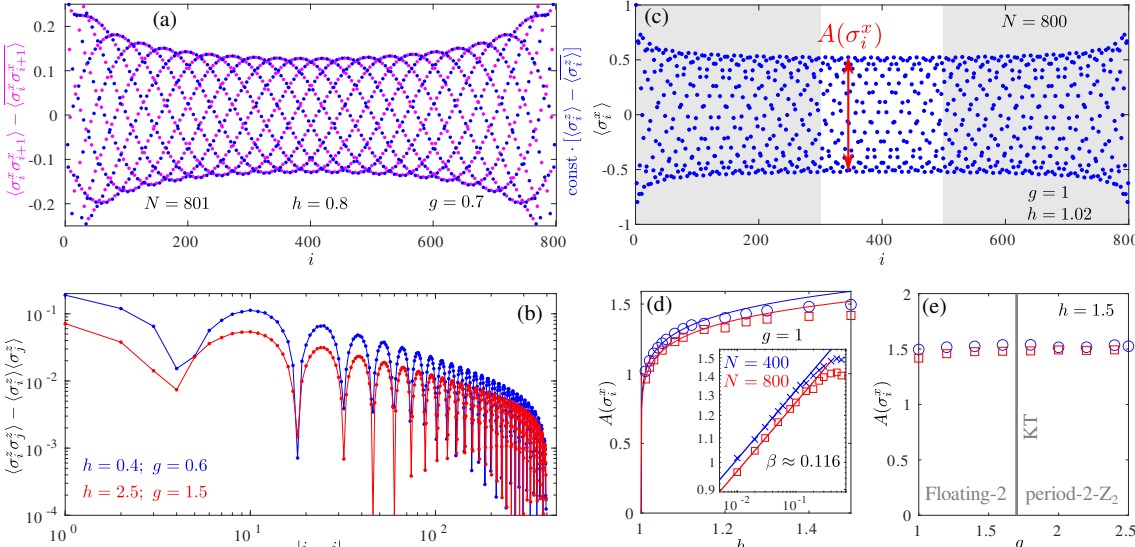

Figure 5: (a) Friedel oscillations of the local magnetization $\sigma_i^z$ (blue) and of the pairing operator $\sigma_i^x \sigma_{i+1}^x$ (magenta) inside the floating-1 phase ($h = 0.8$, $g = 0.7$). Both have been centered around their mean values, the former has been re-scaled with a non-universal pre-factor const $\approx 0.84$. (b) Decay of the magnetization correlations $\langle \sigma_i^z \sigma_j^z \rangle - \langle \sigma_i^z \rangle \langle \sigma_j^z \rangle$ as a function of distance at the two dual points inside the Floating-1 (blue) and Floating-2 (red). (c) Local magnetization along $x$ direction inside the Floating-2 phase with boundaries $x$-polarized. The amplitude of the oscillations $A(\sigma_i^x)$ is extracted as a difference between the largest and smallest values that $\langle \sigma_i^x \rangle$ takes over an interval of length $N/4$ in the middle of the chain. (d) The amplitude $A(\sigma_i^x)$, extracted as in panel (c) along an horizontal cut at $g = 1$, is plotted upon approaching the Ising transition at $h = 1$. Extracted critical exponent $\beta \approx 0.116$ agree within 8% with the theory prediction for Ising transition $\beta = 1/8$. Inset: the same plot in a log-log scale. (e) The amplitude $A(\sigma_i^x)$ along vertical cut at $h = 1.5$ across the Kosterlitz-Thouless transition between Floating-2 and period-2-$\mathbb{Z}_2$.

the correlations $\langle \sigma_i^z \sigma_j^z \rangle - \langle \sigma_i^z \rangle \langle \sigma_j^z \rangle$ for a given point in the Floating-1 $(h, g)$ and the pairing correlations $\langle \sigma_i^x \sigma_{i+1}^x \sigma_j^x \sigma_{j+1}^x \rangle - \langle \sigma_i^x \sigma_{i+1}^x \rangle \langle \sigma_j^x \sigma_{j+1}^x \rangle$ in its dual point in Floating-2 $(1/h, g/h)$, and vice versa. But it turns out that the connection is even stronger since at the same point, these two operators: local magnetization $\sigma_i^z$ and pairing $\sigma_i^x \sigma_{i+1}^x$ have the same scaling dimension $K$. This can be observed by comparing the profiles of the Friedel oscillations presented in Fig. 5 (a). This connection also manifests itself in the same decay of the correlations, for instance $\langle \sigma_i^z \sigma_j^z \rangle - \langle \sigma_i^z \rangle \langle \sigma_j^z \rangle$ at the two dual points as shown in Fig. 5 (b). In both cases there is a non-universal pre-factor in the correlators, see Fig. 5 (a-b).

What distinguishes the two floating phases is the $\mathbb{Z}_2$ symmetry broken in the Floating-2 phase. In order to demonstrate it we look at the order parameter that we associate with an amplitude of the $x$-component of the magnetization $A(\sigma_i^x)$. To detect Ising transition at small $g$ one traditionally define the order parameter as $|\langle \sigma_i^x - \sigma_{i+1}^x \rangle|$ (with $x$-polarized boundaries, see Appendix D). Here we have to adjust it to make it suitable for incommensurate correlations. This is done by computing the maximal and minimal values of the $x$ component of the local magnetization in the middle of the chain and over an interval equal to the quarter of the chain. The example is shown in Fig. 5 (c). Next, we keep track of the amplitude $A(\sigma_i^x)$ upon approaching the transition at $h = 1$, by fitting the data to $\propto (h - 1)^\beta$ we extract the corresponding critical exponent $\beta \approx 0.116$. Our results are in a reasonable agreement with $\beta = 1/8$ for the Ising transition. As mentioned above the pairing operator is relevant for

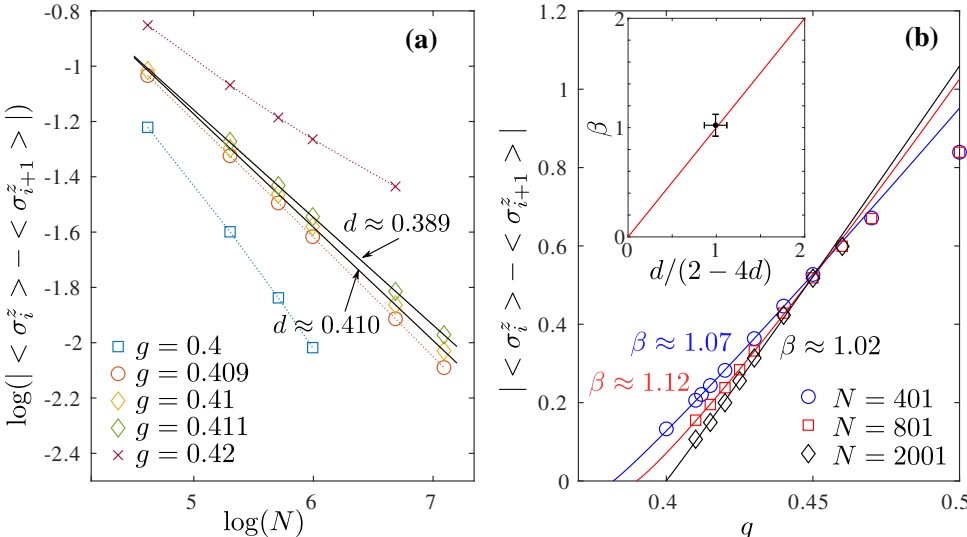

Figure 6: Identification of the eight-vertex universality class. (a) Finite-size scaling of the amplitude of oscillations in the local magnetization. Separatrix is associated with the critical point, the slope corresponds to the scaling dimension $d$. (a) Upper and lower bounds are identified. (b) Scaling of the order parameter with the distance to the critical point. Symbols are DMRG data, lines are fits $\propto (g - g^c)^\beta$. Inset: Comparison between the extracted values of $\beta$ and $d$ for the interacting Majorana chain (black point) with Baxter's eight-vertex model prediction Eq. (7) (red line).

$K > 1/2$. Thus for $h > J$ the pairing that appears in the Floating-2 and period-2-$\mathbb{Z}_2$ phases never constitute a relevant perturbations and remains insensitive to the Kosterlitz-Thouless transition at large-$g$ as shown in Fig. 5 (e).

## 3.3 The eight-vertex point

At $h = 0$ the floating phase collapses into a single point, thus the transition between the period-2 and the $\mathbb{Z}_2$ phase becomes direct. When written in terms of Dirac fermions, the model of Eq. (2) obeys particle-hole symmetry along $h = 0$. Based on the previous study the transition between the $\mathbb{Z}_2$ and the period-2 phases along the particle-hole symmetry line is expected to be in the universality class of the eight-vertex model [15, 41, 50, 51].

According to Baxter [53] all critical exponents of the eight vertex model depend on a single parameter $\rho$:[3]

$$\nu = \pi/(2\rho), \quad \beta = (\pi - \rho)/(4\rho). \tag{6}$$

In special cases, e.g. for the integrable model, the parameter $\rho$ can be expressed in therms of coupling constants [51, 53], however in general this is not possible. Nevertheless, the special relation between $\nu$ and $\beta$ should hold for any value of $\rho$. It means that we can express one exponent in terms of another, say $\beta = \beta(\nu)$ excluding $\rho$ from the equation. In practice, however, numerical estimate of the scaling dimension $d = \beta/\nu$ is way more accurate than the estimate of the correlation length critical exponent $\nu$. Note that by contrast to $\beta$ and $\nu$ that regulates the order parameter and the correlation length as a function of distance to the transition, the scaling dimension $d$ can be extracted at the critical point and reflect the dependence on the system size. After a simple algebra, we got the following prediction for the eight vertex model:

$$\beta = d/(2 - 4d). \tag{7}$$

---

[3]In Baxter's original notations this control parameter was called $\mu$.

In order to verify this prediction numerically we extract independently $\beta$ and $d$. We extract the scaling dimension $d$ of the local order parameter that in the present case is given by the alternation between local magnetization on even and odd sites $|\langle \sigma_i^z \rangle - \langle \sigma_{i+1}^z \rangle|$ - vanishing in the $\mathbb{Z}_2$ phase and finite in the period-2 one. The critical point is associated with the separatrix in a log-log scale and according to Fig. 6 (a) is located in the interval $0.41 < g^c < 0.411$. The slope corresponds to the scaling dimension and lies in the range $0.389 \lesssim d \lesssim 0.41$.

We measure the second critical exponent $\beta$ by looking how alternation of the local magnetization vanishes upon approaching the transition. The results are presented in Fig. 6 (b). Note that the shift of the location of the critical point with system size is typical and has been observed in the previous study of the eight-vertex point [41]. But the critical exponent $\beta$ changes very little with the system size and we take the maximal difference as an estimate for the errorbar. The agreement between obtained numerical results and the theory prediction of Eq. (7) is spectacular and is presented in the inset of Fig. 6 (b). Note that there is no fitting or adjustment parameter and here the comparison between the numerics and the theory is direct.

## 4   $h = 1$ line

The $h = 1$ line is self-dual, and it hosts very rich critical behaviors. The exploration of the extended phases away from this line helps us to get deeper insights into the criticalities realized at $h = 1$. Although our numerical results in general agree with the numerical results provided in Ref. [17] ( see the Appendix E for more comparison), by exploring the surroundings of the critical line we arrived at a different conclusion regarding the nature of the critical phases along $h = 1$ for large $g$. Below we provide a detailed overview of this very challenging line.

### 4.1   The boundaries of the Luttinger liquid

Let us come back to the original arguments regarding the stability of the floating phase. For the operator that simultaneously create $p$ domain walls the scaling dimension is given by $p^2/4K$, the operator becomes relevant when the corresponding scaling dimension is smaller than 2. This implies that the floating phase is stable against superconducting instability of the form $c_i^\dagger c_{i+1}^\dagger + $ h.c. when $2^2/4K > 2$, in other words, for $K < 1/2$. Similar argument applies in the paramagnetic phase: an elementary excitation in this case is a spin flip that cannot create a single domain wall but only a pair of them. Therefore, the transition to the $\mathbb{Z}_2$ or to the paramagnetic phase takes place when the Luttinger liquid exponent reaches its critical value $K^c = 1/2$ [19]. However, these arguments are not valid along $h = 1$ critical line because none of the two operators become relevant - the theory remains critical and the parity symmetry is not broken along this line. We thus associate the transition along $h = 1$ with the point where $K$ reaches the value $K = 1$ of free-fermion theory.

We extract the Luttinger liquid parameter $K$ by fitting the profile of the Friedel oscillations. We polarize edge spins in $z$ direction and fit the data as shown in Fig. 3. Note that the Ising transition with the scaling dimension $d = 1/8$ is controlled by different operator that also requires different boundary conditions (see for instance Appendix D). According to our data presented in Fig. 7 (a) the Luttinger parameter reaches its critical value $K^c = 1$ at $g \approx 0.29$, in good agreement with Ref. [17]. At the same point we observe that the central charge jumps from $c \approx 1/2$ typical for the Ising transition to $c \approx 3/2$ as shown in Fig. 7 (c). Around the same point, the incommensurability appears, therefore the multicritical point M is also a Lifshitz point. In Fig.fig:h1(d) we show that the lowest excitation energies scale to zero as $1/N^z$ with the dynamical critical exponent of the Lifshitz point $z = 3$. All these results agree with the previous study by Rahmani *et al.* [17].

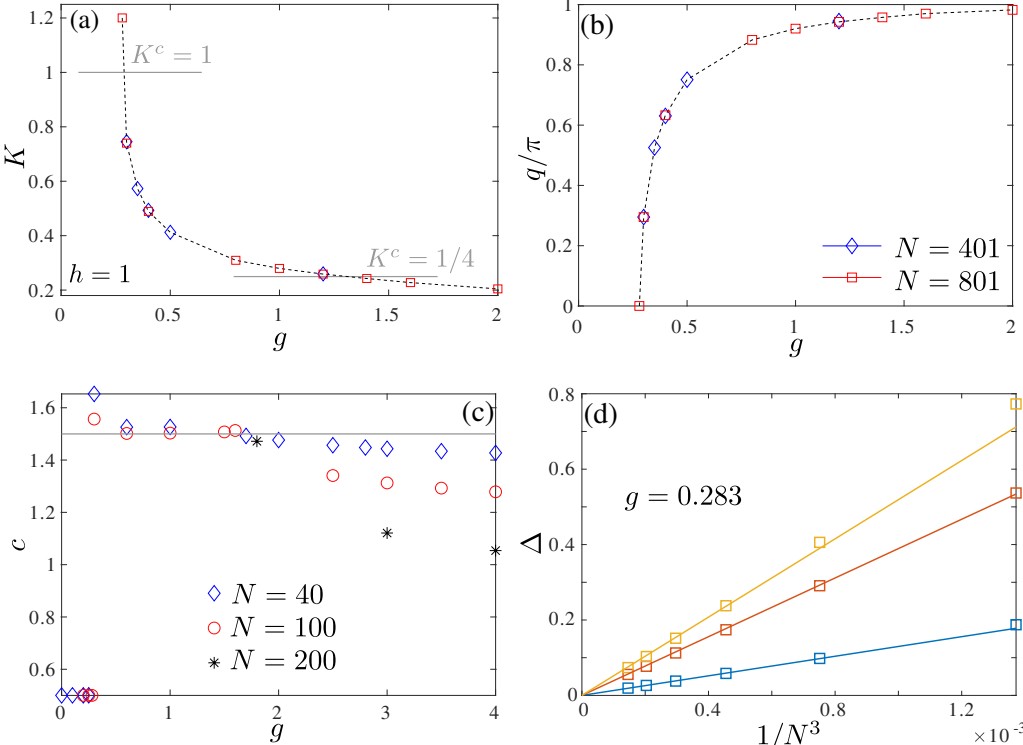

Figure 7: (a) Luttinger liquid parameter $K$ and (b) incommensurate wave-vector $q$ extracted by fitting the Friedel oscillations along $h = 1$ line with open and fixed boundary conditions. (c) Central charge as a function of coupling constant $g$ extracted from the entanglement entropy for chains with periodic boundary conditions. (d) Finite-size scaling of the three lowest excitation energies at the multicritical point M for systems with up to 19 sites with open and fixed boundary conditions. The scaling is in excelent agreement with the theory prediction $\Delta \propto N^{-z}$ for the Lifshitz point with $z = 3$.

When does the Luttinger liquid terminate? For large values of $g$ the operator that becomes relevant is the one that breaks translation symmetry and lead to the period-2 phases. This happens for $K < 1/4$ [19]. By contrast to the low-$g$ case, however, the translation symmetry is broken in both surrounding gapped phases and at the critical line at $h = 1$ between them. So the critical value of the Luttinger liquid exponent is always equal to $K^c = 1/4$. According to our data presented in Fig. 7 (a), along $h = 1$ the Luttinger liquid parameter reaches the critical value $K^c = 1/4$ around $g \approx 1.3$.

Note that the central charge extracted from entanglement entropy in periodic system starts to deviate from $c = 3/2$ for large $g$. As one can see in Fig. 7 (c) this deviation is very slow (as expected for a Kosterlitz-Thouless transition) but it is also very systematic. According to our finite-size data for the entanglement entropy (restricted to small lengths due to periodic boundary conditions) the deviation starts around $g \approx 1.8$. This overestimates the end of the Luttinger liquid compared to the more reliable numerical estimates of Luttinger parameter. In Ref. [17] such deviation starts in the interval $0.5 < g^{-1} < 0.75$ (this corresponds to $1.33 < g < 2$). In both cases the deviation from the $c = 3/2$ starts much earlier than the expected end point at $g \approx 3$ ($g \approx 2.86$ in Ref. [17]) where the transition turns into $1^{\text{st}}$ order.

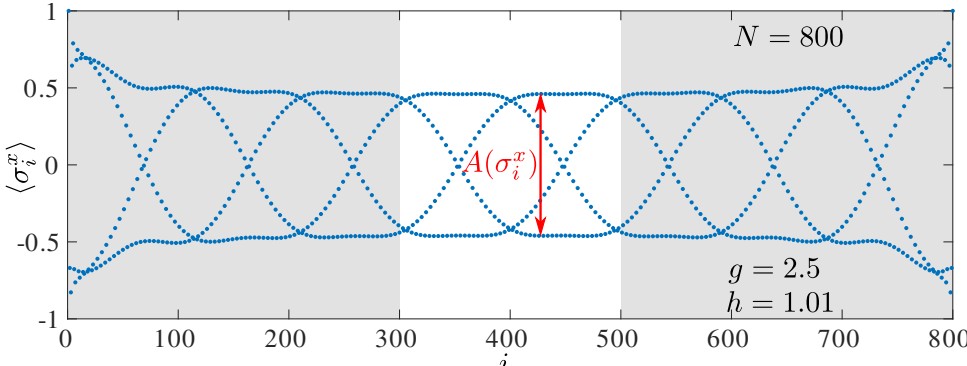

Figure 8: Local magnetization along $x$ direction inside the period-2-$\mathbb{Z}_2$ phase. The amplitude of the oscillations $A(\sigma_i^x)$ is extracted as a difference between the largest and smallest values that $\langle\sigma_i^x\rangle$ takes over an interval of length $N/4$ in the middle of the chain. Presented results are for $N = 800$ site with boundaries polarized in the $x$ direction.

## 4.2 Continuation of the Ising transition and its end point

What happens along $h = 1$ line after the Luttinger liquid terminates? For larger values of $g$ we expect the transition to continue in the Ising universality class in agreement with spontaneously broken $\mathbb{Z}_2$ symmetry between the two gapped phases. In order to confirm this prediction numerically we extract the critical exponent $\beta$ by fitting the order parameter upon approaching the transition. As discussed above in order to adjust the Ising order parameter to the incommensurate case we consider an amplitude of the x-component of the magnetization $A(\sigma_i^x)$ over the finite window in the middle of the chain. An example for the period-2-$\mathbb{Z}_2$ phase is shown in Fig. 8. We keep track of the amplitude $A(\sigma_i^x)$ when approaching the transition from the period-2-$\mathbb{Z}_2$ phase, by fitting the data we extract the corresponding critical exponent $\beta$. In Fig. 9 we show that along $g = 2$ and $g = 2.5$ the amplitude vanishes with the critical exponent that is in excellent agreement with the prediction $\beta = 1/8$ for Ising critical line. According to our results the end point of the Ising critical line is located around $g \approx 3$ (which is within the errorbar of the end point predicted in Ref. [17]). At this point the extracted critical exponent $\beta \approx 0.0413$ agrees within 1% with the CFT prediction $\beta = 1/24$ for the tri-critical Ising point.

Beyond this end point the amplitude remains finite (see Fig. 9) indicating the first order transition between the phases. The possibility of the first order transition in an extended version of the phase diagram has been already discussed in Rahmani *et al.* [17] but it was suggested that the ground-state is four-fold degenerate along this line. We think that the degeneracy is actually higher - six-fold - this corresponds to the level crossings of the two-fold-degenerate state from the period-2 phase and four-fold degenerate one from the period-2-$\mathbb{Z}_2$ phase. Six-fold degeneracy, or in other words three-fold degeneracy in each sector of broken translation would also agree with the tri-critical end point detected numerically.

## 4.3 The persistency of incommensurability

So far we have shown that the Luttinger liquid phase terminates at the point which is different from the end point of the Ising critical line. Let us now show that at none of these two points the incommensurability terminates. In Fig. 10 we show the scaling of the spin-spin correlation with distance, where on top of the main decay one can clearly distinguish the helices typical for incommensurate correlations. Note that on $N = 801$ the presence of incommensurability is still clearly visible for $g = 3.5$. We expect that the correlations slowly approach but probably never reach the commensurate value of $q$ for any finite $g$.

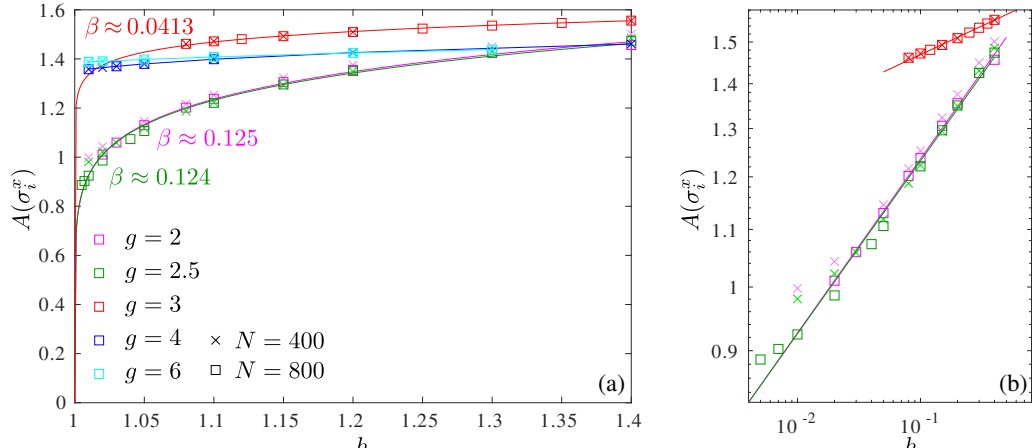

Figure 9: Scaling of the amplitude $A(\sigma_i^x)$ upon approaching the transition between the period-2 and the period-2-$\mathbb{Z}_2$ phase. Symbols are DMRG data ; red, magenta and green lines are fits of the results for $N = 800$ to the power-law $A(\sigma_i^x) \propto (h-1)^\beta$; blue and cyan lines are guide to the eyes. Along $g = 2$ and $g = 2.5$ the extracted exponent $\beta$ is in excellent agreement with the theory prediction for Ising critical exponent $\beta = 1/8$, while for $g = 3$ the critical exponent agrees with the tri-critical Ising end point with $\beta = 1/24$. For $g = 4$ and $g = 6$ the amplitude remains finite up to $h = 1$ in agreement with a first order transition. Inset shows the same data in log-log scale.

# 5 Majorana zero modes

## 5.1 Definition and properties of strong Majorana zero modes

The spontaneous breaking of $\mathbb{Z}_2$ symmetry, commonly signaled by a local (magnetic) order in the thermodynamic limit, may take a non-local form in terms of fermions, as first observed by Kitaev [2] for non-interacting chains. Indeed, in the ordered phase of the TFI chain (see Eq. (1) with $g_x = g_z = 0$ and $J > h$), fermions display non-trivial topological properties, with "unpaired" zero-energy Majorana edge states localized at the boundaries of an open chain. This brought the notion of so-called strong zero-mode (SZM), popularized by Fendley [54]. This terminology "strong" refers to the fact that the entire many-body spectrum is concerned, in contrast with "weak" zero modes which only touch the low-energy part [55,56]. We recall that a SZM operator $\Psi_{\mathrm{zm}}$ must have the three following properties:

1) Normalizable: $\Psi_{\mathrm{zm}}^\dagger \Psi_{\mathrm{zm}} = 1$.

2) Commuting with the Hamiltonian (at least in the thermodynamic limit) $[\mathcal{H}, \Psi_{\mathrm{zm}}] \to 0$.

3) Anti-commuting with the discrete symmetry: $\{\mathbb{P}, \Psi_{\mathrm{zm}}\} = 0$, where $\mathbb{P} = \prod_i \sigma_i^z$ is the parity operator.

A striking consequence of these properties is that $\Psi_{\mathrm{zm}}$ provides a map between even and odd sectors for the entire many-body spectrum of $\mathcal{H}$ which therefore displays a pairing degeneracy that becomes exact in the thermodynamic limit. For a finite system, this pairing degeneracy is usually lifted and a small residual finite-size parity gap exists between even and odd sectors

$$\Delta_{\mathrm{p}}(N) = \left| E_{\mathrm{odd}} - E_{\mathrm{even}} \right| \sim \exp\left(-\frac{N}{\xi_{\mathrm{zm}}}\right), \tag{8}$$

exponentially vanishing with a correlation length $\xi_{\mathrm{zm}}$ which also controls the edge state localization [2, 54]. Below we discuss Majorana zero modes physics, first for non-interacting systems where SZM operators can be exactly constructed. This will help us to understand

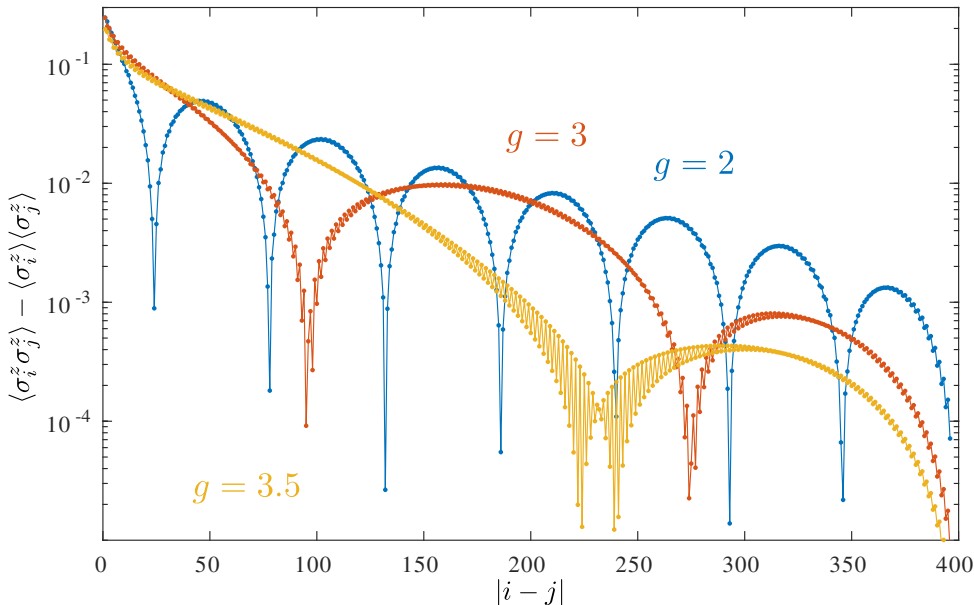

Figure 10: Correlations as a function of distance for $h = 1$ and and various values of $g$. The incommensurability persists even beyond the tri-critical end point at $g \approx 3$.

the behavior of the parity gap in the $\mathbb{Z}_2$ ground-state of the interacting problem at small $g$, a regime well captured by a self-consistent mean-field decoupling.

## 5.2 Non-interacting Kitaev chains

### 5.2.1 Transverse-field Ising chain

There are only a few cases where exact expressions for SZM operators $\Psi_{zm}$ are known, the simplest and most famous example being the TFI chain model, defined for an open wire of $N$ sites by

$$\mathcal{H} = J \sum_{j=1}^{N-1} \sigma_j^x \sigma_{j+1}^x - h \sum_{j=1}^{N} \sigma_j^z = -\mathrm{i}\left( J \sum_{j=1}^{N-1} b_j a_{j+1} - h \sum_{j=1}^{N} a_j b_j \right). \tag{9}$$

With open boundary conditions, one can easily express the SZM operators, localized at the edges

$$\Psi_{zm}^{\text{Left}} = \sum_{j=1}^{N} \Theta_j a_j \quad \text{and} \quad \Psi_{zm}^{\text{Right}} = \sum_{j=1}^{N} \Theta_j b_{N+1-j}. \tag{10}$$

The amplitude $\Theta_j$ decays exponentially away from each open end

$$\Theta_j \propto \exp\left(-\frac{j}{\xi_{zm}}\right), \quad \text{with} \quad \xi_{zm} = \frac{1}{\ln(J/h)}. \tag{11}$$

### 5.2.2 Incommensurability and parity switches for Kitaev wires

One can also derive exact expressions in the more general case provided by the non-interacting lattice model of p-wave superconducting wires [2], when hopping and pairing terms are not necessarily equal $t \neq \Delta$. This model can be written using the three equivalent languages

(spins, Majorana and Dirac fermions: see Appendix A.2), as follows

$$
\begin{aligned}
\mathcal{H} &= \sum_j \left[ t\left( c_j^\dagger c_{j+1} + \text{h.c.} \right) + \Delta\left( c_j^\dagger c_{j+1}^\dagger + \text{h.c.} \right) + 2hn_j \right] \\
&= \sum_j \left[ X\sigma_j^x \sigma_{j+1}^x + Y\sigma_j^y \sigma_{j+1}^y - h\sigma_j^z \right] \\
&= -\mathrm{i} \sum_j \left[ X b_j a_{j+1} - Y a_j b_{j+1} - h a_j b_j \right],
\end{aligned} \tag{12}
$$

where $t = X + Y$ and $\Delta = X - Y$. The condition of existence for SZM is quite simple (see Appendix B) as it simply requires $X + Y > h$. However, as first discussed for spin correlations in their seminal work by Barouch and McCoy [57], there is an interesting oscillatory regime for the pair-wise correlations where precisely the SZM also display incommensurate (IC) modulation [58,59]. This occurs under the simple condition $h^2 < 4XY$, with an IC wave-vector $q$ given by

$$
\cos q = \frac{h}{2\sqrt{XY}} . \tag{13}
$$

The amplitude $\Theta_j$ of the left and right zero-modes operators Eq. (10) then obeys

$$
\Theta_j \propto \sin(qj) \exp\left( -\frac{j}{\xi_{\mathrm{zm}}^{\mathrm{IC}}} \right), \quad \text{with} \quad \frac{1}{\xi_{\mathrm{zm}}^{\mathrm{IC}}} = \ln\sqrt{\frac{X}{Y}} . \tag{14}
$$

In such a case, one can show that the finite-size parity gap also displays some IC modulation. It is straightforward to obtain an exact analytical expression for the parity gap (see Appendix B.2 for details), which in the incommensurate regime reads

$$
\Delta_{\mathrm{p}}^{\mathrm{(IC)}}(N) = 2X \left( M_x^{\mathrm{s}} \right)^2 \frac{\sin\left[ q(N+1) \right]}{\sin q} \exp\left( -\frac{N}{\xi_{\mathrm{zm}}^{\mathrm{IC}}} \right), \tag{15}
$$

where $M_x^{\mathrm{s}}$ is the surface magnetization (see Appendix B.2). Contrary to the TFI model, here the finite-size parity gap may vanish exactly if $\sin\left[ q(N+1) \right] = 0$ and $q \neq 0$. On a finite chain of length $N$, this occurs $N/2$ times, i.e. when

$$
\frac{h_n^*}{2\sqrt{XY}} = \cos\left( \frac{n\pi}{N+1} \right), \quad n = 1, \ldots, \frac{N}{2}, \tag{16}
$$

corresponding to a level crossing within each doublet of opposite parity $p = \pm 1$, associated with a parity switch [59]. The commensurate-incommensurate crossover inside the $\mathbb{Z}_2$ phase occurs for $n = 0$, i.e. when $h_{\mathrm{IC}} = 2\sqrt{XY}$. On the other side (commensurate regime for $X + Y > h > h_{\mathrm{IC}}$) one recovers a monotonous exponential decay of the form Eq. (8) with

$$
\frac{1}{\xi_{\mathrm{zm}}^{\mathrm{C}}} = \ln\left( \frac{X}{h + \sqrt{h^2 - 4XY}} \right). \tag{17}
$$

The SZM localization length $\xi_{\mathrm{zm}}$ and the incommensurate vector $q$, given by Eqs. (13), (14) and (17), are both shown in Fig. 11 (a) for the non-interacting Kitaev chain model Eq. (12) with hopping $t = 1$ and pairing $\Delta = 0.8$ ($X = 0.9$, $Y = 0.1$) as a function of the field $h$. The surface magnetization $M_x^{\mathrm{s}}$ (which plays the role of the order parameter of the $\mathbb{Z}_2$ phase, see Appendix B.2) is also plotted. In panel (b) we show for the same set of parameters how the parity gap $\left| E_{\mathrm{odd}} - E_{\mathrm{even}} \right|$ behaves as a function of $h$. For various system lengths, $\Delta_{\mathrm{p}}$ oscillates in the incommensurate regime (with $N/2$ oscillations), as predicted.

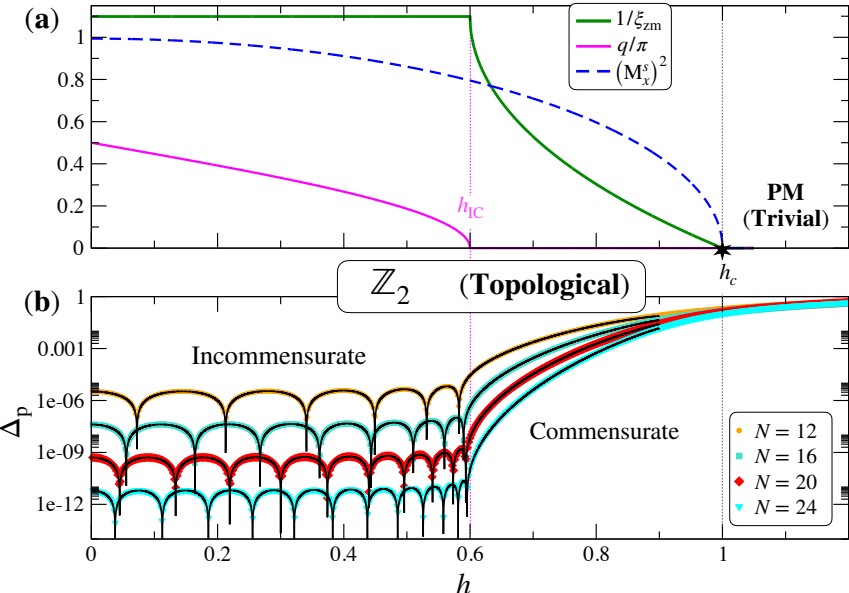

Figure 11: Free-fermion results for the non-interacting Kitaev chain model Eq. (12) with open boundary conditions, hopping $t = 1$ and pairing $\Delta = 0.8$ ($X = 0.9$, $Y = 0.1$). (a) Field $h$-dependence of the SZM localization length $\xi_{zm}$ and the incommensurate vector $q$, given by Eqs. (13), (14) and (17), plotted together with the surface magnetization $M_x^s$ (see Appendix B.2). (b) The finite-size parity gap $\Delta_p$ (symbols: exact diagonalization results ; lines: analytical expressions, see Appendix B.3) displays incommensurate oscillations, as predicted by Eq. (15) for $h < 2\sqrt{XY} = 0.6$, followed by a commensurate $\mathbb{Z}_2$ topological phase, and then the PM (trivial) regime for $h > 1$.

## 5.3 Interacting case: DMRG results

In contrast with free-fermion systems, the situation is much more complicated in the interacting case [13, 14]. Despite recent efforts addressing the very existence of possible SZM operators in the presence of interaction [60–63], no explicit and exact construction of SZM is known, except the notable exemple of the integrable XYZ chain [64]. For the interacting and non-integrable model Eq. (1) with $g_x \neq g_z \neq 0$, Kemp $et\ al.$ [38] have shown the existence of an "almost" SZM that almost commutes with the Hamiltonian. Nevertheless, evidences for edge modes have been provided for various interacting models, but only at low energy. Interestingly, incommensurability may help this detection with the possibility to continuously tune the effective coupling (and therefore the parity gap) between the edges [7, 65, 66], and eventually make it vanishing. However, the strong character of the SZMs turns out to be much more difficult to grasp because it addresses the whole many-body spectrum, as recently discussed in the context of many-body localization at infinite temperature [31, 67]. Here we will focus on ground-state physics, and therefore will not address this issue.

### 5.3.1 $\mathbb{Z}_2$ antiferromagnet at small $g$

We first numerically investigate the effect of weak interaction strength $g$ on the Majorana zero-modes of the TFI chain model in the $\mathbb{Z}_2$ regime $h < 1$ using DMRG simulations, addressing the spectral pairing of the two lowest energy levels of the many-body spectrum. We detect the presence of zero modes by computing finite-size energy difference between the two ground-states with open boundary conditions. This is done by targeting multiple states of the effective Hamiltonian and keeping track of the energy as a function of DMRG iterations. The method

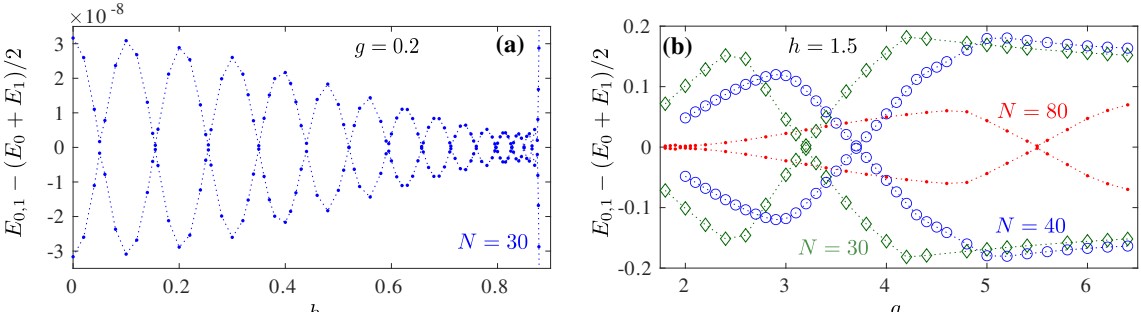

Figure 12: DMRG results for the ground-state zero modes of $\mathbb{Z}_2$ phases of the interacting Majorana chain model Eq. (4). $E_0$ states for the ground-state energy and $E_1$ is the energy of the in-gap excitation. (a) Weak interaction $\mathbb{Z}_2$ phase: relative energy of the in-gap states as a function of field $h$ along an horizontal cut at $g = 0.2$ computed for $N = 30$ sites. Incommensurate correlations tuned by $h$ lead to exact level crossings between the two in-gap states with even and odd parity. (b) Strongly interacting regime $\mathbb{Z}_2$-period-2 phase: same as (a) but as a function of $g$ for $h = 1.5$. Exact crossings (also induced by incommensurability) are clearly visible, together with an overall decay with increasing system sizes $N = 30, 40, 80$.

is described in details in Ref. [49].

In Fig. 12 (a) we present results for the parity gap $\Delta_p$ along an horizontal cut in the phase diagram (Fig. 2) at $g = 0.2$ as a function of $h$. Results, shown for $N = 30$ sites, are displayed with respect to the average energy of the two lowest levels, such that exact level crossings are clearly visible. Note that similar oscillations (and level crossings) have been observed in Ref. [39] with small chains for the interacting spin model Eq. (1) with $g_x = 0$ and $g_z > 0$. This is a nice example of field-induced parity switches, as already discussed above for the IC regime of the non-interacting Kitaev chain, see Fig. 11 (b) and Eq. (15). In this exactly solvable case, the parity switch condition Eq. (16) yielded $N/2$ level crossings. In the present interacting case, our DMRG results strongly suggest a similar condition with $N/2$ parity switches, as observed numerically in Fig. 12 (a). This conjecture is also supported by the self-consistent MF approach at weak interaction, as we will explain below in Section 5.4.

This exact level crossing regime with parity switches is dictated by the incommensurability vector $q$ on top of the AF $\mathbb{Z}_2$ order. Upon increasing further $h$, this oscillatory regime disappears and is replaced by the more conventional commensurate $\mathbb{Z}_2$ topological order with a monotonous parity gap, but still exponentially vanishing with $N$ as in Eq. (8) ; this occurs for $h > h_{\mathrm{IC}}$ with $h_{\mathrm{IC}} \approx 0.87$ for $g = 0.2$. Below in Fig. 14 we also discuss similar effects for other values of the parameters.

### 5.3.2 $\mathbb{Z}_2$ period-2 antiferromagnet at large $g$

Majorana zero-modes are also expected for the upper right part of the phase diagram (see Fig. 2) in the strong interaction regime for $h > 1$. Indeed, we expect two discrete symmetries to be spontaneously broken there: the translation and the $\mathbb{Z}_2$. A sketchy representation of the 4 associated ground-states would be for the two sectors of translation

$$| \rightarrow \rightarrow \leftarrow \leftarrow \rightarrow \rightarrow \leftarrow \cdots \rangle \pm | \leftarrow \leftarrow \rightarrow \rightarrow \leftarrow \leftarrow \rightarrow \cdots \rangle \tag{18}$$

$$| \leftarrow \rightarrow \rightarrow \leftarrow \leftarrow \rightarrow \rightarrow \cdots \rangle \pm | \rightarrow \leftarrow \leftarrow \rightarrow \rightarrow \leftarrow \leftarrow \cdots \rangle . \tag{19}$$

Although the translation is explicitly broken in our DMRG simulations on open chains, one can still probe the breaking of $\mathbb{Z}_2$ by monitoring the parity gap. Panel (b) of Fig. 12 provides



Figure 13: Schematic picture for the MF Majorana Hamiltonian where the interaction terms now correspond to 2-body Majorana couplings at distance 3. The MF model becomes self-dual when $Y = \widetilde{X}$, which precisely occurs if $X = h$.

DMRG results in this strong interaction regime for $h = 1.5$. One sees exact crossings induced by the incommensurability. Due to proximity of the wave-vector to its commensurate value (at the Kosterlitz-Thouless transition the wave-vector is $q \gtrsim 0.95\pi$) we can detect only one crossing for $N = 30$ and $N = 40$ and two crossings for $N = 80$. As expected the location of the crossing changes with the system size. Of course, one might expect more crossings for larger systems but the accuracy of our simulations are not sufficient to resolve in-gap states in this part of the phase diagram for longer chains. Qualitatively our results are consistent with exponential decay of the amplitude of the energy difference with the system-size $N$.

## 5.4 Self-consistent mean-field theory

We now present a self-consistent mean-field (MF) treatment of the interacting terms which gives surprisingly good results. This approach allows us to gain some physical insight and a better understanding of the interacting $\mathbb{Z}_2$ topological regime, at least for weak interactions.

### 5.4.1 Mean-field Hamiltonian

We perform a MF decoupling of the $g$ interaction terms (see Appendix C) in the weakly interacting limit. This leads to the following effective Hamiltonian in terms of Dirac and Majorana fermions (up to irrelevant constant terms)

$$
\begin{aligned}
\mathcal{H}_{\text{MF}} &= \sum_j \left[ t_j \left( c_j^\dagger c_{j+1} + \text{h.c.} \right) + \Delta_j \left( c_j^\dagger c_{j+1}^\dagger + \text{h.c.} \right) + 2h_j n_j + \widetilde{X}_j \left( c_j^\dagger c_{j+2} + c_j^\dagger c_{j+2}^\dagger + \text{h.c.} \right) \right] \\
&= -\mathrm{i} \sum_j \left[ X_j b_j a_{j+1} - Y_j a_j b_{j+1} - h_j a_j b_j + \widetilde{X}_j b_j a_{j+2} \right],
\end{aligned} \tag{20}
$$

where $t_j = X_j + Y_j$ and $\Delta_j = X_j - Y_j$. Rewriting the same MF Hamiltonian in the spin language is also useful:

$$
\mathcal{H}_{\text{MF}} = \sum_j \left[ X_j \sigma_j^x \sigma_{j+1}^x + Y_j \sigma_j^y \sigma_{j+1}^y - h_j \sigma_j^z + \widetilde{X}_j \sigma_j^x \sigma_{j+1}^z \sigma_{j+2}^x \right]. \tag{21}
$$

Interestingly, the non-interacting MF Hamiltonian is very close to a non-interacting Kitaev chain model Eq. (12), with an additional second neighbor hopping term $-\mathrm{i}\widetilde{X}_j b_j a_{j+2}$ which takes the form of an $\alpha$-chain model [68] with $\alpha = 2$.[4] A sketch of the MF model is given in Fig. 13.

### 5.4.2 Numerical results

The new couplings $\left\{ X_j ; Y_j ; h_j ; \widetilde{X}_j \right\}$ of the MF Hamiltonian Eqs. (20) and (21) are obtained from the self-consistent equations (see Eqs. (C.4 ) in Appendix C where some technical details about the microscopic and convergence aspects are also discussed, OBC are used for the MF

---

[4]Also equivalent to the so-called cluster model in spin language [69,70].

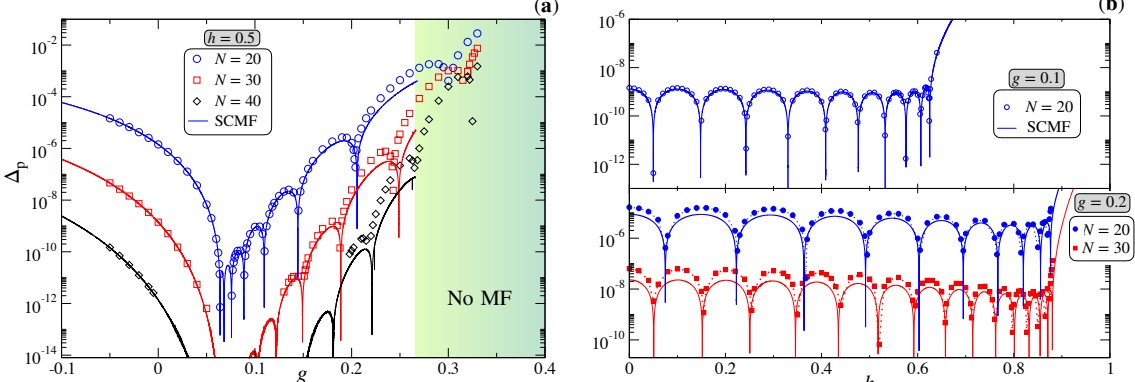

Figure 14: Quantitative comparison between DMRG (symbols) and self-consistent mean-field (SCMF) (lines) for the parity gap $\Delta_p$. (a) Vertical scan in the phase diagram of Fig. 2 at $h = 0.5$: data are shown for $N = 20, 30, 40$. The agreement for $g$ not too large is quite remarkable, the green shaded area signals the region where no MF convergence can be obtained. (b) Horizontal scan at $g = 0.1$ (upper panel) and $g = 0.2$ (bottom panel): the agreement, quantitatively excellent for $g = 0.1$, slightly deteriorates as $g$ increases, but the periodicity of the oscillations remains perfectly captured by the SCMF approach.

parameters). One can easily reach a converged MF solution at weak interaction $g$, which is the target regime of such Hartree-Fock decouplings, but we also observe a breakdown of the MF convergence when $g$ exceeds a certain value $\sim 0.25$, see below in Section 5.4.3 where the MF phase diagram is shown.

$\mathcal{H}_{\mathrm{MF}}$ is numerically diagonalized iteratively till the convergence of the MF parameters $\left\{X_j; Y_j; h_j; \widetilde{X}_j\right\}$. In Fig. 14, we illustrate the exact diagonalization results of the converged Hamiltonian $\mathcal{H}_{\mathrm{MF}}$ for the parity gap $\Delta_p$. A direct comparison with DMRG calculations is provided for the evolution of $\Delta_p$ (a) *vs.* $g$ for $h = 0.5$, and (b) against $h$ for $g = 0.1$ and $0.2$. The agreement is quantitatively excellent for weak interaction strength, typically below $\sim 0.2$, with both the amplitudes and the incommensurate oscillations very well captured despite the approximation. Interestingly the parity gap displays a similar behavior as compared to the standard non-interacting Kitaev chain (Fig. 11), which undoubtedly captures the essential features of this $\mathbb{Z}_2$ regime of the phase diagram.

### 5.4.3 Mean-field phase diagram

One can further gain insights about the qualitative effects produced by the interactions on the non-interacting $\mathbb{Z}_2$ line of the TFI model, by looking more closely to the MF decoupling in Eqs. (C.4 ). Indeed, keeping only the dominant corrections, we approximate the new couplings by

$$X_j \approx J - g\left(|\mathcal{C}_{j-1}^{xx}| + |\mathcal{C}_{j+1}^{xx}|\right), \quad Y_j \approx g|\mathcal{C}_j^{xx}|, \quad h_j \approx h - g\left(m_{j-1}^z + m_{j+1}^z\right), \quad \widetilde{X}_j \approx g m_{j+1}^z, \quad (22)$$

where $m_j^z = \langle \sigma_j^z \rangle$ and $C_j^{xx} = \langle \sigma_j^x \sigma_{j+1}^x \rangle$. These expressions can be even more simplified in the limit of small field $h$, where $m_j^z$ is small and $C_j^{xx} \approx -1$ (AF order). Assuming uniform couplings, the toy-model with $\left\{X_j; Y_j; h_j; \widetilde{X}_j\right\}_{\forall j} = \left\{J - 2g; g; h; 0\right\}$, exactly solvable, is expected to capture some features of the interacting Majorana chain model in the $\mathbb{Z}_2$ regime at small enough interaction $g$ and field $h$.

In Fig. 15 (a) we verify that the disorder line separating commensurate and incommensurate regions of the $\mathbb{Z}_2$ phase, trivially given by $h_{\mathrm{IC}}^2 = 4g$ for the toy-model, reproduces very well

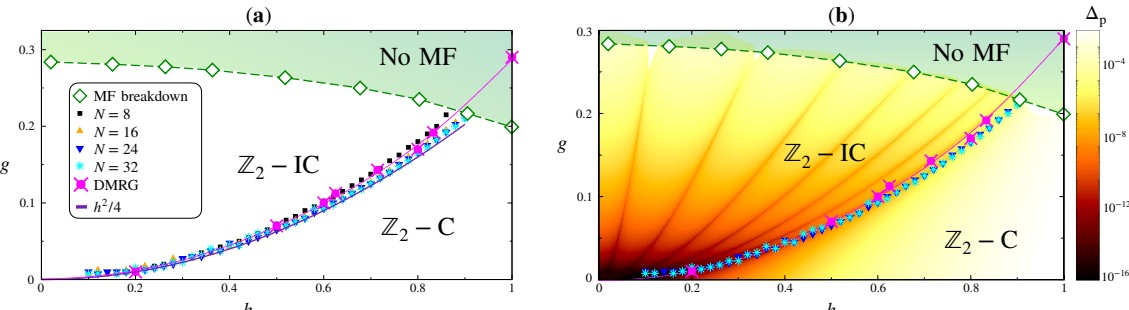

Figure 15: SCMF phase diagram in the weak-interaction regime displayed in the plane field $h$ – interaction $g$. (a) The C-IC disorder line, computed using SCMF for various chain lengths (as indicated on the plot), is found in excellent agreement with DMRG data (the magenta line is a guide to the eyes).The full line $g = h^2/4$ is the toy-model estimate for the C-IC line (see text) which gives an excellent description. Green diamonds show the boundary of the SCMF convergence: in the top green area SCMF does not converge anymore. (b) Color map of the parity gap $\Delta_p$, numerically obtained for $N = 16$. The parity switches lines are clearly visible where the parity gap vanishes.

the more involved SCMF results. Quite remarkably, this remains true for the entire $\mathbb{Z}_2$ regime, well beyond the $h \ll 1$ limit. A direct comparison with DMRG results for the commensurate-incommensurate line is also is excellent agreement.

The right panel of Fig. 15 (b) provides the MF phase diagram of $\mathcal{H}_{\mathrm{MF}}$ as a color map of the parity gap in the plane field $h$ – interaction $g$. In addition to the disorder line, one sees $N/2$ dark lines showing the minima of $\Delta_{\mathrm{p}}$ which signal $N/2$ parity switches. For readability and clarity, this is shown for a small system size $N = 16$.

# 6 Discussions and conclusions

In this work we have presented an extended phase diagram of the interacting Majorana chain in the case of symmetric interaction $g_z = g_x$, as overviewed in Fig. 2. The obtained phase diagram is very rich and contains four gapped phases and two floating phases, i.e. Luttinger liquid regimes with incommensurate correlations. In addition, our results motivated us to revisit the nature of the effective critical theories along the self-dual critical line at $h = 1$. We argue that there is no generalized commensurate-incommensurate transition where Ising and Luttinger liquid criticalities both terminate. We demonstrate that the Luttinger liquid phase stops at $g \approx 1.3$, while the Ising critical line persists beyond it up to $g \approx 3$ where it ends with a tri-critical Ising point. Incommensurability persists beyond these two points and, if it ends, it does so for a much stronger $g$ interaction.

We have also provided numerical evidence of a multi-critical point in the eight-vertex universality class at $h = 0$ and $g \approx 0.41$. Note that due to duality one can expect a second critical point in the eight-vertex universality class at $h \to \infty$ (or $J \to 0$) and $g/h \approx 0.41$ between the paramagnetic and the period-2-$\mathbb{Z}_2$ phases. This implies that both, translation and parity, symmetries will be broken on one side of the transition and, quite surprisingly this does not change the nature of the critical point.

Interestingly, the floating phases in the present phase diagram are separated from the phases with broken translation symmetry by Kosterlitz-Thouless transitions, by contrast to previous studies where a Pokrovsky-Talapov transition has been reported [19, 41]. This is be-

cause the short-range order in the gapped phases is characterized by incommensurate wave-vectors, and thus the transition between floating and period-2 phases are incommensurate-incommensurate (and not commensurate-incommensurate as in the case of Pokrovsky-Talapov transition). This short-range incommensurability is realized when $g_z = g_x$ and absent when $g_x = 0$. Therefore by tuning $g_x/g_z$ one might expect a crossover between the Pokrovsky-Talapov and the Kosterlitz-Thouless transitions similar to the one reported recently in a chain of spinless fermions with next-nearest-neighbor interactions [71]. It would be interesting to check this prediction numerically.

The tri-critical Ising conformal field theory is supersymmetric, thus we might expect that the end point of the Ising critical line will have the features of the supersymetric critical point, for example, doubly degenerate excitation spectra. But the tri-critical Ising point is not the only place in this phase diagram where one might expect supersymmetry to appear. The Ising critical line that lies within the gapless region with U(1) and $\mathbb{Z}_2$ symmetry might have an emergent $\mathcal{N} = (1, 1)$ supersymmetry [72, 73]. In the present case, the $\mathbb{Z}_2$ is satisfied by the Hamiltonian, while the U(1) symmetry is an emergent symmetry stabilizing the Luttinger liquid phase [17, 19]. Moreover, at the multicritical point where Ising transition exit the Luttinger liquid phase and hits the Kosterlitz-Thouless transition one might expect the spontaneously emergent $\mathcal{N} = (3, 3)$ supersymmetry. According to Ref. [74] this higher supersymmetry can be realized under the following three conditions: *i)* the system has to be invariant with respect to U(1) and $\mathbb{Z}_2$ symmetry ; *ii)* it has to be tuned to the multicritical point where Ising and Kosterlitz-Thouless transitions coincide ; and *iii)* the velocity of the fermionic degree of freedom should be smaller than or equal to the velocity of the bosonic degree of freedom. The first two conditions are formally satisfied at the both multicritical points terminating the $c = 3/2$ line (see points M and S at the phase diagram of Fig. 2). Although these two points appears very differently: point S appears as an intersection between the Ising the Kosterlitz-Thouless lines; while due to continuity of the equal-K lines the Kosterlitz-Thouless transition is expected to approach the point M at an infinite slope. The last condition on the velocities requires further investigation. We leave this problem for future studies.

Finally, we have explored the topological properties of the systems by looking at the possible existence of Majorana zero-modes, signalled by the (two-fold) parity degeneracy. The $\mathbb{Z}_2$ topological order of the (Kitaev) free-fermion (for $h < 1$) remains robust against weak interactions, as unambiguously shown by large-scale DMRG simulations, and remarkably captured by a self-consistent fermionic mean-field theory. Due to incommensurability, one observes a succession of exact zero-mode level crossings already for finite chain lengths. A similar observation is also reported at large $g$ in the $\mathbb{Z}_2$ period-2 regime. Despite such good evidences for the existence of Majorana edge modes at low energy, their existence at higher energies, as it is the case in the absence of interaction, remains highly hypothetical.

# Acknowledgements

NC acknowledges useful discussions with Naoki Kawashima and Kareljan Schoutens. NL is grateful to Pasquale Marra for stimulating discussions. This work has been supported by Delft Technology Fellowship (NC) and by the French National Research Agency (NL), project GLADYS ANR-19-CE30-0013, and (NL) the EUR grant NanoX No. ANR-17-EURE-0009 in the framework of the "Programme des Investissements d'Avenir". Numerical simulations have been performed at the Dutch national e-infrastructure with the support of the SURF Cooperative.

## A  Useful transformations

### A.1  Duality

The model defined by the Hamiltonian Eq. (1) up to boundary terms transforms into itself by Kramers-Wannier duality transformation:

$$\sigma_i^x \sigma_{i+1}^x \rightarrow \tau_i^z \quad \text{and} \quad \sigma_i^z \rightarrow \tau_i^x \tau_{i+1}^x, \tag{A.1}$$

where $\sigma$ and $\tau$ are Pauli matrices. Then the duality transformation is given by:

$$\mathcal{H} = \sum_i \sigma_i^x \sigma_{i+1}^x + h \sum_i \sigma_i^z + g \sum_i \left( \sigma_i^x (\sigma_{i+1}^x)^2 \sigma_{i+2}^x + \sigma_i^z \sigma_{i+1}^z \right)$$

$$= h \left[ \sum_i \frac{1}{h} \tau_i^z + \sum_i \tau_i^x \tau_{i+1}^x + \frac{g}{h} \sum_i (\tau_i^z \tau_{i+1}^z + \tau_i^x \tau_{i+2}^x) \right], \tag{A.2}$$

where we used that $(\sigma_{i+1}^x)^2 = \mathbb{I}$ and the fact that the model is symmetric with respect to the sign of the field $h$. Up to a pre-factor the Hamiltonian of Eq. (A.2) is equivalent to the original Hamiltonian in Eq. (1) with $h \rightarrow h^{-1}$ and $g \rightarrow g/h$. This duality transformation allows us to study the nature of the quantum phase transitions only for $h \leq 1$, and re-use these results for $h > 1$.

### A.2  Jordan-Wigner and Majorana fermions

The Jordan-Wigner transformation maps Pauli operators to Dirac fermions

$$\sigma_j^z = 1 - 2c_j^\dagger c_j, \tag{A.3}$$

$$\sigma_j^x = K_j \left( c_j^\dagger + c_j \right), \tag{A.4}$$

$$\sigma_j^y = \mathrm{i} K_j \left( c_j^\dagger - c_j \right), \tag{A.5}$$

$$\text{with} \quad K_j = \prod_{k=1}^{j-1} \sigma_k^z. \tag{A.6}$$

Therefore the original spin model Eq. (1) can be rewritten as an interacting fermions problem, see Eq. (2). One can also introduce two Majorana fermion operators $a_j$ and $b_j$

$$a_j = c_j^\dagger + c_j = K_j \sigma_j^x,$$
$$b_j = \mathrm{i}(c_j^\dagger - c_j) = K_j \sigma_j^y,$$
$$a_j b_j = \mathrm{i}(1 - 2c_j^\dagger c_j) = \mathrm{i}\sigma_j^z,$$

which satisfy the usual rules: $(a/b)^\dagger = (a/b)$, $(a/b)^2 = 1$, $\{a_i, a_j\} = \{b_i, b_j\} = 2\delta_{ij}$, and $\{a_i, b_j\} = 0$. Then the interacting problem can be re-written in terms of Majorana variables as given by the Hamiltonian Eq. (3).

## B  Strong zero modes for the non-interacting Kitaev model

### B.1  SZM construction: iterative procedure

In the Majorana representation, the Kitaev model reads

$$\mathcal{H} = -\mathrm{i} \sum_j \left[ X b_j a_{j+1} - Y a_j b_{j+1} - h a_j b_j \right]. \tag{B.1}$$

Assuming simple linear combinations for the (normalized) SZM operators

$$\Psi_{\text{left}} = \frac{1}{\mathcal{N}_N} \sum_{j=1}^{N} \Theta_j \, a_j \quad \text{and} \quad \Psi_{\text{right}} = \frac{1}{\mathcal{N}_N} \sum_{j=1}^{N} \Theta_j \, b_{N+1-j} \,, \tag{B.2}$$

and using $\left[\mathcal{H}, a_j\right] = 2\mathrm{i}\left(X b_{j-1} - h b_j + Y b_{j+1}\right)$ and $\left[\mathcal{H}, b_j\right] = 2\mathrm{i}\left(-Y a_{j-1} + h a_j - X a_{j+1}\right)$, we iteratively arrive at the simple recursion relation for $\Theta_j$:

$$\Theta_{j+1} = \frac{h}{X} \Theta_j - \frac{Y}{X} \Theta_{j-1} \,, \tag{B.3}$$

such that

$$\left[\mathcal{H}, \Psi_{\text{left}}\right] = 2\mathrm{i} \frac{1}{\mathcal{N}_N} \left(Y \Theta_{N-1} - h \Theta_N\right) b_N \quad \text{and} \quad \left[\mathcal{H}, \Psi_{\text{right}}\right] = 2\mathrm{i} \frac{1}{\mathcal{N}_N} \left(Y \Theta_{N-1} - h \Theta_N\right) a_1 \,. \tag{B.4}$$

It is easy to solve the above recursion Eq. (B.3) with initial conditions $\Theta_0 = 0$ and $\Theta_1 = 1$. We restrict the present discussion to positive couplings $h, X, Y \geq 0$ and $X \geq Y$ (it is straightforward to get results for the other cases).

The phase diagram of the Kitaev chain model can simply be inferred from the existence of normalizable SZM, which requires $h < X + Y$. Contrary to the TFIM case, here the topological regime is richer as one can distinguish two types of SZM decays.

**(i) Commensurate regime if $h^2 \geq 4XY$.**

$$\Theta_j = \frac{X}{\alpha h} \left(\frac{h}{2X}\right)^j \times \left[(1+\alpha)^j - (1-\alpha)^j\right] \xrightarrow[j \gg 1]{} \begin{cases} \infty \,, & \text{if } h > X + Y \,, \\ \frac{X}{\alpha h} e^{-j/\xi_{\text{zm}}^{\text{C}}} \,, & \text{if } X + Y > h > 2\sqrt{XY} \,, \end{cases}$$

where $\alpha = \sqrt{1 - 4XY/h^2}$, and the edge mode localization length given by

$$\frac{1}{\xi_{\text{zm}}^{\text{C}}} = \ln\left[\frac{2X}{(1+\alpha)h}\right] \,. \tag{B.5}$$

Note that in the limit $\alpha \to 0$ ($h \to 2\sqrt{XY}$ from above)

$$\Theta_j \xrightarrow[\alpha \to 0]{} j \left(\frac{h}{2X}\right)^{j-1} \,. \tag{B.6}$$

The SZM normalization factor can be expressed in the large $N$ limit:

$$\frac{1}{\mathcal{N}} \xrightarrow[N \to \infty]{} 2\alpha \left[\frac{1}{(1+\alpha)^{-2} - \left(\frac{h}{2X}\right)^2} + \frac{1}{(1-\alpha)^{-2} - \left(\frac{h}{2X}\right)^2} - \frac{2}{(1-\alpha^2)^{-1} - \left(\frac{h}{2X}\right)^2}\right]^{-1/2} \,. \tag{B.7}$$

**(ii) Incommensurate (IC) regime if $h^2 < 4XY$.**

$$\Theta_j = \frac{2X}{\sqrt{4XY - h^2}} \sin(qj) \, e^{-j/\xi_{\text{zm}}^{\text{IC}}} \tag{B.8}$$

displays oscillations and exponential decay, controlled by

$$\cos q = \frac{h}{2\sqrt{XY}} \quad \text{and} \quad \frac{1}{\xi_{\text{zm}}^{\text{IC}}} = \ln\sqrt{\frac{X}{Y}} \,. \tag{B.9}$$

The normalization factor is given by

$$\frac{1}{\mathcal{N}} \xrightarrow[N \to \infty]{} \sqrt{2} \sin q \left[\frac{1}{1 - \frac{Y}{X}} + \frac{\frac{Y}{X} - \cos(2q)}{1 - 2\frac{Y}{X} \cos(2q) + \left(\frac{Y}{X}\right)^2}\right]^{-1/2} \,. \tag{B.10}$$

### B.2 Surface magnetization

An hallmark of the topological phase is the presence of edge states. The surface magnetization, a key-quantity in that respect, is defined by [75]

$$\mathrm{M}_x^{\mathrm{s}} = \langle \mathcal{E}^- | \sigma_{1,L}^x | \mathcal{E}^+ \rangle, \tag{B.11}$$

where $|\mathcal{E}^\pm\rangle$ are many-body eigenstates of parity $p = \pm$ associated to a partner $|\mathcal{E}^\mp\rangle$ of opposite parity $-p$, obtained by acting the SZM operator, i.e. $\Psi_{\mathrm{left/right}}|\mathcal{E}^\pm\rangle \approx |\mathcal{E}^\mp\rangle$. From the definition of the SZM operators Eq. (B.2), we easily arrive at

$$\mathrm{M}_x^s = \frac{1}{\mathcal{N}}, \tag{B.12}$$

for which we have analytical expressions in both C and IC regimes, see Eqs. (B.7), (B.10).

### B.3 Finite size parity gap and parity switches

The (finite-size) parity gap is obtained from the simple expression

$$
\begin{aligned}
\Delta_{\mathrm{parity}} &= \langle \mathcal{E}^- | \mathcal{H} | \mathcal{E}^- \rangle - \langle \mathcal{E}^+ | \mathcal{H} | \mathcal{E}^+ \rangle \approx \langle \mathcal{E}^- | [\mathcal{H}, \Psi_{\mathrm{zm}}] | \mathcal{E}^+ \rangle \\
&\approx \frac{X}{\mathcal{N}} \Big( \langle \mathcal{E}^- | a_1 | \mathcal{E}^+ \rangle - i \langle \mathcal{E}^- | b_N | \mathcal{E}^+ \rangle \Big) \Big( \frac{h}{X} \Theta_N - \frac{Y}{X} \Theta_{N-1} \Big) \\
&\approx 2X \left( \mathrm{M}_x^{\mathrm{s}} \right)^2 \Big( \frac{h}{X} \Theta_N - \frac{Y}{X} \Theta_{N-1} \Big).
\end{aligned}
\tag{B.13}
$$

We then obtain closed forms for both regimes. In the commensurate case

$$
\begin{aligned}
\Delta_{\mathrm{parity}}^{(\mathrm{C})} &\approx 2X \left( \mathrm{M}_x^{\mathrm{s}} \right)^2 \left( \frac{1+\alpha}{2\alpha} \right) \left( \frac{h}{2X} \right)^N \times \left[ (1+\alpha)^N - (1-\alpha)^N \right] \\
&\approx \begin{cases} 2X \left( \mathrm{M}_x^{\mathrm{s}} \right)^2 \left( \frac{1+\alpha}{2\alpha} \right) \mathrm{e}^{-N/\xi_{\mathrm{zm}}^{\mathrm{C}}}, & \text{if } \alpha \neq 0, \\ 2X \left( \mathrm{M}_x^{\mathrm{s}} \right)^2 \mathrm{e}^{-N/\xi_{\mathrm{zm}}^{\mathrm{C}}} \times N, & \text{if } \alpha \to 0, \end{cases}
\end{aligned}
\tag{B.14}
$$

where $\alpha = \sqrt{1 - 4XY/h^2}$ and $\xi_{\mathrm{zm}}^{\mathrm{C}}$ is given by Eq. (B.5). For the incommensurate case we get

$$\Delta_{\mathrm{parity}}^{(\mathrm{IC})} \approx 2X \left( \mathrm{M}_x^{\mathrm{s}} \right)^2 \frac{\sin[q(N+1)]}{\sin q} \mathrm{e}^{-\frac{N}{\xi_{\mathrm{zm}}^{\mathrm{IC}}}}, \tag{B.15}$$

where $\xi_{\mathrm{zm}}^{\mathrm{IC}}$ and $q$ are given by Eq. (B.9).

## C Self-consistent mean-field

**Mean-field decoupling and self-consistent equations.** We start from the interacting Majorana chain model

$$\mathcal{H} = -i \sum_j \left( J b_j a_{j+1} - h a_j b_j \right) - g \sum_j \left( a_j b_j a_{j+1} b_{j+1} + b_j a_{j+1} b_{j+1} a_{j+2} \right), \tag{C.1}$$

where the quartic term that cannot be treated exactly. We make a mean-field decoupling which leads to a noninteracting system quadratic fermionic problem, with new coupling constants which are self-consistently determined. The quartic terms are decoupled in all mean-field channels which are consistent with the Wick's theorem [39]:

$$\begin{aligned}
a_j b_j a_{j+1} b_{j+1} &\approx a_j b_j \langle a_{j+1} b_{j+1} \rangle + \langle a_j b_j \rangle a_{j+1} b_{j+1} \\
&\quad + a_j b_{j+1} \langle b_j a_{j+1} \rangle + \langle a_j b_{j+1} \rangle b_j a_{j+1}, \\
b_j a_{j+1} b_{j+1} a_{j+2} &\approx b_j a_{j+1} \langle b_{j+1} a_{j+2} \rangle + \langle b_j a_{j+1} \rangle b_{j+1} a_{j+2} \\
&\quad + b_j a_{j+2} \langle a_{j+1} b_{j+1} \rangle + \langle b_j a_{j+2} \rangle a_{j+1} b_{j+1}.
\end{aligned} \tag{C.2}$$

This leads to the following mean-field Hamiltonian (up to irrelevant constant terms)

$$\mathcal{H}_{\mathrm{MF}} = -\mathrm{i} \sum_j \left[ X_j b_j a_{j+1} - Y_j a_j b_{j+1} - h_j a_j b_j + \widetilde{X}_j b_j a_{j+2} \right], \tag{C.3}$$

where the parameters are computed from the following self-consistent equations

$$\begin{aligned}
h_j &= h + \mathrm{i}g \left( \langle a_{j-1} b_{j-1} \rangle + \langle a_{j+1} b_{j+1} \rangle + \langle b_{j-1} a_{j+1} \rangle \right) = h - g \left( m^z_{j-1} + m^z_{j+1} + C^{xzx}_{j-1,j,j+1} \right), \\
X_j &= J - \mathrm{i}g \left( \langle b_{j-1} a_j \rangle + \langle b_{j+1} a_{j+2} \rangle + \langle a_j b_{j+1} \rangle \right) = J + g \left( C^{xx}_{j-1,j} + C^{xx}_{j+1,j+2} - C^{yy}_{j,j+1} \right), \\
Y_j &= \mathrm{i}g \langle b_j a_{j+1} \rangle = -g C^{xx}_{j,j+1}, \\
\widetilde{X}_j &= -\mathrm{i}g \langle a_{j+1} b_{j+1} \rangle = g m^z_{j+1}.
\end{aligned} \tag{C.4}$$

The local magnetization $m^z_j = \langle \sigma^z_j \rangle$, the correlators at distance one (involving two sites) $C^{\beta\beta}_{j,j+1} = \langle \sigma^\beta_j \sigma^\beta_{j+1} \rangle$, and at distance two (involving three sites) $C^{xzx}_{j-1,j,j+1} = \langle \sigma^x_{j-1} \sigma^z_j \sigma^x_{j+1} \rangle$ are computed in the ground-state of $\mathcal{H}_{\mathrm{MF}}$. We use OBC and therefore there is a dependence on the position along the open chain, which dies off exponentially away from the boundaries. However, we have checked that such a dependence is a finite-size effect which does not influence the MF phase diagram. The self-consistent loop is performed numerically, and convergence is obtained when a stationary solution is reached for the coupling parameters in Eq. (C.4).

In Fig. 16 we show the evolution of the effective MF parameters (evaluated in the middle of the open chain for $j = N/2$) as a function of the interaction $g$ for a three values of the transverse field $h = 0.1, 0.5, 0.9$. Several remarks are in order here. Both the transverse magnetic field and the nearest neighbor coupling $X$ decrease with $g$, while in the same time the newly generated terms $Y$ and $\widetilde{X}$ both grow. This can be naturally understood from the SCMF equations Eq. (C.4), which can be approximated by $h_j \approx h - 2gm$, $X_j \approx J + 2g\mathcal{C}$, $Y_j \approx -g\mathcal{C}$, and $\widetilde{X}_j \approx gm$, where in the limit of small field $h \ll J$, the on-site magnetization $m \approx h$ and the nearest-neighbor antiferromagnetic correlator $\mathcal{C} \approx -1$.

**Convergence.**   In panel (d) of Fig. 16 we also show for number of MF iterations $\tau_{\mathrm{MF}}$ required to globally converge all local MF parameters (to a relative precision of $10^{-5}$). $\tau_{\mathrm{MF}}$ gradually increases with $g$, with a notable enhancement at not too large interaction strength, due to tiny parity gap in the system, but MF still converge (at least provided the parity gap does not reach machine precision). However there is a clear breakdown of MF theory at larger $g$ where $\tau_{\mathrm{MF}}$ diverges (vertical arrows).

# D   Ising transition for small $g$

In the main text we extracted Luttinger liquid exponent $K$ by fitting the local magnetization in the $z$ direction that in terms of bosons/fermions corresponds to the local density. For this we fix the boundary condition to be polarized along $z$. However, in order to detect Ising transition breaking $\mathbb{Z}_2$ symmetry we have to look at the different local operator - $\langle \sigma^x_i - \sigma^x_{i+1} \rangle$. It is very

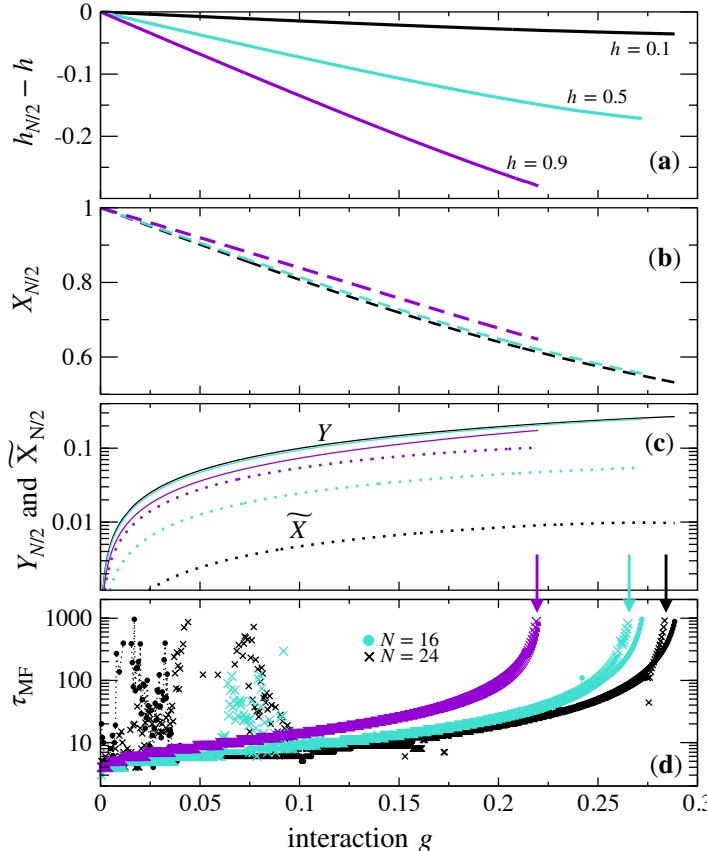

Figure 16: Mean-field parameters obtained after numerically solving the SCMF equations Eq. (C.4) for $h = 0.1, 0.5, 0.9$ plotted against $g$.

intuitive to see why this term plays a role of an order parameter for the Ising transition: for small $g$ and $h$ the ground-state corresponds to the antiferromagnetic alignment of spins along $x$-direction, thus the absolute difference between the two projection is large in the $\mathbb{Z}_2$ phase, while it vanishes in the paramagnetic phase. Important to notice, that in order to use this local operator one has to fix the boundary conditions by applying the boundary field along $x$. In order to keep the profile symmetric we apply the same boundary field at each edge and take odd number of sites $N = 801$. The results are presented in Fig. 17. From the DMRG data (blue and green) one can see that the difference between the profiles at $g = 0$ and upon approaching the floating phase $g = 0.25$ is negligible. In both cases fit to the theory prediction of the profile gives the estimate of the scaling dimension $d \approx 0.126$ which is in excellent agreement with the CFT prediction for the Ising model $d = 1/8$.

# E   Comparison to the previous study at $h = 1$

In the main text we mentioned that although we arrived to a different conclusions regarding critical regimes along $h = 1$ line our numerical results to a large extent agrees with the previous study by Rahmani et al [17]. In Fig. 18 we compare our results for the Luttinger liquid parameter $K$ extracted in the present paper by fitting the Friedel oscillations and those computed from the finite-size energy spectra by Rahmani et al. [17]. Overall we see a very good agreement between the results. There is slight discrepancy on where the Luttinger liquid

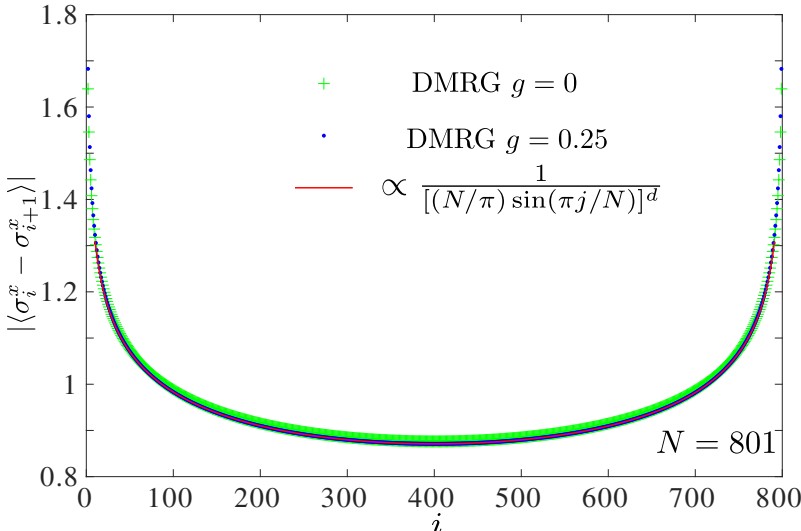

Figure 17: Fridel oscillation profile at the Ising transition at $h = 1$ and $g = 0$ (green) and $g = 0.25$ (blue). Red line is the result of the fit with $d \approx 0.126$.

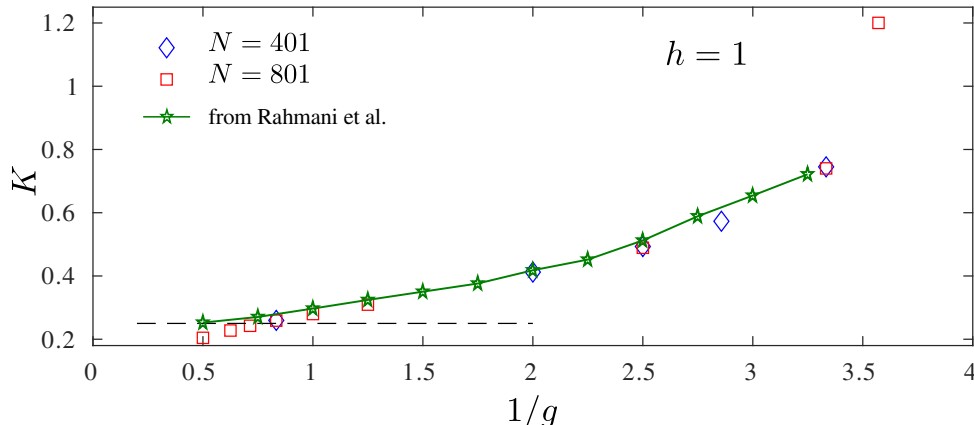

Figure 18: Comparison between the Luttinger liquid parameter $K$ extracted in this paper with Friedel oscillations (blue and red) versus Luttinger liquid parameter extracted frpm the plots presented in Ref. [17] and obtained with energy of the low-lying excitations.

parameter crosses 1/4 line: according to our data this happens around $g \approx 1.3$ ($1/g \approx 0.75$), according to Rahmani et al's data, this happens at $1/g \approx 0.5$ or $g = 2$. The main source of this discrepancy is probably associated the finite-size effects in the energy spectra computed on systems with up to 200 Majoranas (100 spins) and used in Ref. [17] to extract Luttinger liquid exponent $K$. But in any case, according to all available data $K$ reaches its critical value 1/4 well below the end point of the continuous Ising transition located at $g \approx 3$.

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
