# Peer review of "Topological and quantum critical properties of the interacting Majorana chain model"

_SciPost Physics, doi:SciPost Phys. 14, 152 (2023)_

## Round 1 · Referee Report · Dirk Schuricht · 2023-1-4

Report

The article studies the phase diagram of the Majorana chain with self-dual interactions. The bulk of the results are obtained using numerical simulations. The presentation is very clear, including a well-plotted phase diagram in Fig. 2. The connection to previous results, in particular Refs. 17 and 19, are clearly discussed. The same is true for the remaining open points, thus opening the pathway for future research. In summary the work is interesting and well-suited for publication (apart from a few minor points, see list below). Thus I recommend publication in SciPost.

I have some questions or suggestions which should be considered before publication:
-in Sec. 2.1.1, what is meant with "zero modes vanishing exponentially"? does it refer to exponentially localised?
-is it possible to write down an effective field theory for the floating phases, eg, via bosonisation, or has this already been discussed in the literature?
-is it known what happens to the $c=3/2$ critical line if the condition $g_x=g_z$ is relaxed?
-it is stated that the results on the point M are in agreement with Ref. 17, does this also concern the dynamical critical exponent $z=3$ reported there?
-are the DMRG results reported in Fig. 15 for the full model or for the SCMF model?
-some references seem misplaced, are doubled (see Ref. 35), or miss journal information (eg, Ref. 56)

  • validity: -
  • significance: -
  • originality: -
  • clarity: -
  • formatting: -
  • grammar: -

Author:  Nicolas Laflorencie  on 2023-02-03  [id 3310]

(in reply to Report 1 by Dirk Schuricht on 2023-01-04)
Category:
answer to question

---

## Round 1 · Referee Report · Dirk Schuricht · 2023-1-4

The article studies the phase diagram of the Majorana chain with self-dual interactions. The bulk of the results are obtained using numerical simulations. The presentation is very clear, including a well-plotted phase diagram in Fig. 2. The connection to previous results, in particular Refs. 17 and 19, are clearly discussed. The same is true for the remaining open points, thus opening the pathway for future research. In summary the work is interesting and well-suited for publication (apart from a few minor points, see list below). Thus I recommend publication in SciPost.
I have some questions or suggestions which should be considered before publication:

Our response to Report 1

-> We are glad and acknowledge Dirk Schuricht for this positive report. Below we address all his points.

-in Sec. 2.1.1, what is meant with "zero modes vanishing exponentially"? does it refer to exponentially localised?

-> Indeed, we expect such an effect, while we do not have a simple expression for the strong zero mode operators. Nevertheless, we clearly observe an exponentially vanishing parity gap.
Changes in the manuscript: Therefore we have rephrased “zero modes vanishing exponentially with a system size” onto “zero modes showing a vanishing (parity) gap, exponentially suppressed with the system size.” in the new version of the manuscript

-is it possible to write down an effective field theory for the floating phases, eg, via bosonisation, or has this already been discussed in the literature?

-> Floating phase is an incommensurate Luttinger liquid phase with an emergent U(1) symmetry so one can use all the existing machinery there. Quite extensive study of the stability of the floating phases against superconducting and density-wave instabilities has been done by Verresen, Pollmann, Vishwanath see https://arxiv.org/abs/1903.09179 (our Ref.[19])

-is it known what happens to the c=3/2 critical line if the condition gx=gz is relaxed?

-> When gx is different from gz the model is no longer self-dual. Let’s imagine the 3D phase diagram where an additional direction will be, say gx/gz. None of these two terms, gx or gz, can destroy immediately the floating phases. This is also true (and quite obvious) for the four gapped phases. It means that away (and at least not too far) from the the plane gx=gz the nature of the transition has to be the same - in the Ising between the pairs of the gapped phases and between the two floating phases. But, since there is no duality for gx different from gz, there is no reason for the Ising transition to take place at h=1. The most general case with 3D parameter space is, of course, tricky to explore, and it goes far beyond the scope of the present work.

-it is stated that the results on the point M are in agreement with Ref. 17, does this also concern the dynamical critical exponent z=3 reported there?

-> We have carefully explored this question of the value of the dynamical exponent , which was not numerically proved in Ref. [17]. We have performed additional exact diagonalization simulations and have now clear evidence for , as shown in Fig. 7 (d) for the so-called M point. The finite size scaling of the first three gaps above the ground-state nicely show a behaviour, thus confirming .
Changes in the manuscript: We have generated a new Fig. 7 with a panel (d) showing this finite size scaling of the first low-energy gaps.

-are the DMRG results reported in Fig. 15 for the full model or for the SCMF model?

-> The DMRG points in the SCMF phase diagram are for the full interacting odel. This makes the SCMF solution quantitatively correct!

-some references seem misplaced, are doubled (see Ref. 35), or miss journal information (eg, Ref. 56)
-> Thank you very much for such a careful reading. We have fixed this small issue.
Changes in the manuscript: Note that we have merged [15] and [35], and added [24] to the group [20-25].
[24] M. McGinley, J. Knolle and A. Nunnenkamp, Robustness of majorana edge modes and topological order: Exact results for the symmetric interacting Kitaev chain with disorder, Phys. Rev. B 96, 241113 (2017), doi:10.1103/PhysRevB.96.241113.

---

## Round 1 · Referee Report · Anonymous · 2023-1-19

Report

Dear Editor,

the article "Topological and quantum critical properties of the interacting Majorana chain model" deals with the study of the ground-state phase diagram of an interacting Majorana chain, or, which is almost equivalent, of a modified quantum Ising model.
The article is impressive for the thoroughness of the study, the impressive amount of high-quality numerical simulation and the clarity of the explanation. The phase diagram in Fig. 2 will likely be a milestone reference for future studies.
The work surely deserves publication in Scipost.

I have a few minor comments.

- I would ask the authors to state more clearly that topological phases are strictly speaking "topological" only if the problem is looked at from the fermionic viewpoint. If one works with the spin-1/2 Ising model, these are just magnetically-ordered phases. I understand completely that numerical simulations are easily carried out in the spin language but that at the some time one would like to speak also about the fermionic model, but it is important to warn the unexperienced reader.

- Is it possible to gain some insights in the phase diagram by studying the limiting lines, e.g. g=infty? Is g=0 the only exactly-solvable line of the model?

- in the abstract: hoping --> hopping

  • validity: -
  • significance: -
  • originality: -
  • clarity: -
  • formatting: -
  • grammar: -

Author:  Nicolas Laflorencie  on 2023-02-03  [id 3309]

(in reply to Report 2 on 2023-01-19)
Category:
answer to question

---

## Round 1 · Referee Report · Anonymous · 2023-1-19

Dear Editor,
the article "Topological and quantum critical properties of the interacting Majorana chain model" deals with the study of the ground-state phase diagram of an interacting Majorana chain, or, which is almost equivalent, of a modified quantum Ising model.
The article is impressive for the thoroughness of the study, the impressive amount of high-quality numerical simulation and the clarity of the explanation. The phase diagram in Fig. 2 will likely be a milestone reference for future studies. The work surely deserves publication in Scipost.

-> We are very pleased to read such a positive review, and acknowledge the Referee. Below we address all points.

I have a few minor comments.
-I would ask the authors to state more clearly that topological phases are strictly speaking "topological" only if the problem is looked at from the fermionic viewpoint. If one works with the spin-1/2 Ising model, these are just magnetically-ordered phases. I understand completely that numerical simulations are easily carried out in the spin language but that at the some time one would like to speak also about the fermionic model, but it is important to warn the unexperienced reader.

-> This is indeed a crucial point which worth some attention in particular when we go back and forth between both (magnetic and fermionic) languages. We have added a sentence at the beginning of the paper, when we first describe the different regimes.
Changes in the manuscript: In 2.1.1 we have added: “Note that this phase has no topological interest for the (Ising) spin degrees of freedom for which the so-called "topological phase" there boils down to a more conventional magnetic order.”

-Is it possible to gain some insights in the phase diagram by studying the limiting lines, e.g. g=infty? Is g=0 the only exactly-solvable line of the model?
-> As far as we know, the limit is not solvable, see the discussion in Rhamani et al. [17], below their Eq. (2.9).

-in the abstract: hoping --> hopping
->Corrected.

---

## Round 1 · Referee Report · Anonymous · 2023-1-22

Report

The article „Topological and quantum critical properties of the interacting
Majorana chain model“ studies the phase diagram of an extended quantum Ising model with next nearest neighbor interactions that can be mapped onto an interacting Majorana chain model. The results are well presented and all necessary points are discussed to understand the full phase diagram from Fig. 2. In conclusion the work is very interesting and, therefore, I recommend this manuscript for publication in SciPost.

Some minor recommendations:
- One should discuss more clearly the difference between the fermionic and the spin picture; although they can be mapped onto each other there are some crucial differences
- In Fig. 12 the label of the y-axis, i.e. E_0, E_1and E_{0,1}, is neither mentioned in the main text nor the caption of the figure. I assume that these are the energies of the ingap states, but a clear definition in the caption or the main text is missing.
- In the discussion section „commensurate-incommendurate“ should be „commensurate-incommensurate“

  • validity: -
  • significance: -
  • originality: -
  • clarity: -
  • formatting: -
  • grammar: -

Author:  Nicolas Laflorencie  on 2023-02-03  [id 3308]

(in reply to Report 3 on 2023-01-22)
Category:
answer to question

---

## Round 1 · Referee Report · Anonymous · 2023-1-22

-> We are very pleased to read such a positive review, and acknowledge the Referee. Below we address all points.

Some minor recommendations: - One should discuss more clearly the difference between the fermionic and the spin picture; although they can be mapped onto each other there are some crucial differences

-> This comment goes in the same direction as the one raised above by Referee 2. We reproduce the answer below. This is indeed a crucial point which worth some attention in particular when we go back and forth between both (magnetic and fermionic) languages. We have added a sentence at the beginning of the paper, when we first describe the different regimes. Changes in the manuscript: In 2.1.1 we have added: “Note that this phase has no topological interest for the (Ising) spin degrees of freedom for which the so-called "topological phase" there boils down to a more conventional magnetic order.”

  • In Fig. 12 the label of the y-axis, i.e. E_0, E_1and E_{0,1}, is neither mentioned in the main text nor the caption of the figure. I assume that these are the energies of the ingap states, but a clear definition in the caption or the main text is missing. -> This has been fixed. Changes in the manuscript: We have added a sentence in the caption of Fig. 12 “$E_0$ states for the ground-state energy and $E_1$ is the energy of the in-gap excitation.”

  • In the discussion section „commensurate-incommendurate“ should be „commensurate-incommensurate“ ->This has been fixed.

---

## Round 2 · Author Response

Dear editor,

We are pleased to see that our manuscript has three positive reports and that we are only asked for minor revisions. Please find enclosed our response to all comments, as well as a summary of changes, following some referee suggestions.

Sincerely,
Natalia Chepiga and Nicolas Laflorencie

---

## Round 2 · List of Changes

1/ We have rephrased “zero modes vanishing exponentially with a system size” onto “zero modes showing a vanishing (parity) gap, exponentially suppressed with the system size.”

2/ We have generated a new Fig. 7 with a panel (d) showing the N^{-3} finite size scaling of the first low-energy gaps at the M point.

3/ We have merged [15] and [35], and added [24] to the group [20-25]. [24] M. McGinley, J. Knolle and A. Nunnenkamp, Robustness of majorana edge modes and topological order: Exact results for the symmetric interacting Kitaev chain with disorder, Phys. Rev. B 96, 241113 (2017), doi:10.1103/PhysRevB.96.241113.

4/ In section 2.1.1 we have added: “Note that this phase has no topological interest for the (Ising) spin degrees of freedom for which the so-called "topological phase" there boils down to a more conventional magnetic order.”

5/ We have added a sentence in the caption of Fig. 12 “$E_0$ states for the ground-state energy and $E_1$ is the energy of the in-gap excitation.”

---

## Editorial Decision

published